# INTERACTIVE SEQUENTIAL GENERATIVE MODELS

## ABSTRACT

Understanding spatiotemporal relationships among several agents is of considerable relevance for many domains. Team sports represent a particularly interesting real-world proving ground since modeling interacting athletes requires capturing highly dynamic and complex agent-agent dependencies in addition to temporal components. However, existing generative methods in this field either entangle all latent factors into a single variable and are thus constrained in practical applicability, or they focus on uncovering interaction structures, which restricts their generative ability. To address this gap, we propose a framework for multiagent trajectories that augments graph-structured sequential generative models with explicit latent social dependencies. First, we derive a novel objective within the variational autoencoder family using a disentangled latent space that aims to encapsulate inherent data traits. Based on the proposed training criterion, we then present a model architecture that unifies insights from neural interaction inference and graph-structured variational recurrent neural networks for generating collective movements while allocating latent information. We validate our model on data from professional soccer and basketball. Our framework not only improves upon existing state-of-the-art approaches in forecasting trajectories, but also infers semantically meaningful representations that can be used in downstream tasks.

## 1 INTRODUCTION

The study of agent behavior governed by temporal and spatial dependencies is of great importance in many different fields, such as autonomous driving (Brown et al., 2020; Rasouli & Tsotsos, 2019), robot navigation (Rudenko et al., 2020), or sports analytics (Tuyls et al., 2021). In particular, accurate detection of implicit causal social structures offers several advantages by removing confounding factors for trajectory forecasting tasks and providing practitioners with interpretable dynamics that can in turn be integrated into downstream decision-making processes or applications.

Modeling the dynamics of multiplayer sports games (Omidshafiei et al., 2022; Le et al., 2017; Yue et al., 2014; Liu et al., 2020) is particularly challenging since accurate trajectory generation in this environment requires capturing highly dynamic and complex underlying modular structures (Makansi et al., 2022). For example, the roles prescribed in a team formation are a poor indicator of the actual behavior observed in a given situation. Moreover, most of the interacting elements inject noise into the forecasting process because they are either irrelevant (e.g., goal keepers) or their influential nature changes as the situation evolves. However, existing methods for modeling sports data rely on graph encoding strategies (Kipf & Welling, 2016; Vaswani et al., 2017) that aggregate social information into only single variables that need to capture all latent stochasticity (Zhan et al., 2019; Yeh et al., 2019; Sun et al., 2019; Omidshafiei et al., 2022).

In recent years, a considerable amount of methods have been proposed that aim to infer interactive components in general multiagent systems via discrete latent variables. These methods are usually formulated as some form of variational autoencoder (Kingma & Welling, 2013; Sohn et al., 2015) that learns latent edge categories of an assumed underlying graph structure (Kipf et al., 2018; Graber & Schwing, 2020; Löwe et al., 2022). However, being the only causal factors specified, the proposed frameworks neglect other potential latent characteristics not originating in mere interactive categories but equally affecting multimodal agent behavior, which limits their generative capacity.

To address previous shortcomings, we propose a novel framework for modeling multiagent trajectory data that enhances existing graph-structured latent variable models by explicitly encoding social

structures in sports games. Since the contemplated spatiotemporal systems are caused by dynamic dependencies among heterogeneous agents, we define this component as a causal graph comprising categorical agent roles and pairwise interactions. Based on the specified generative setting, we then introduce an objective function within the variational autoencoder family and instantiate a concrete architecture for computing the derived training components. Empirically, our model exceeds existing state-of-the-art methods in forecasting trajectories on data from professional soccer and basketball. In addition, we report on extensive quantitative and qualitative analyses wrt. the learned latent variables that show informative properties in generative tasks and downstream applications.

## 2 BACKGROUND

Given data $\mathcal{D} = \{\boldsymbol{x}_{\leq T}^{(i)}\}_{i=1}^N$ consisting of $N$ sequences $\boldsymbol{x}_{\leq T} = [\boldsymbol{x}_1, ..., \boldsymbol{x}_T]$, our goal is to estimate the underying data distribution via maximizing the likelihood of the collected evidence, i.e,. $\max p_\theta(\boldsymbol{x}_{\leq T})$. In practice, $p_\theta(\boldsymbol{x}_{\leq T})$ is often highly multimodal, which complicates direct deployment of MLE. A frequently used modeling paradigm for stochasticity in complex multimodal distributions is introducing latent variables and optimizing the variational lower bound on the maginal log-likelihood (Kingma & Welling, 2013; Rezende et al., 2014; Sohn et al., 2015).

Existing conditional variational models for generating highly-structured sequential data $\boldsymbol{x}_{\leq T}$ usually associate a latent variable $\boldsymbol{z}_1, ..., \boldsymbol{z}_T$ with each timestep of the segment to describe the generative process (Bayer & Osendorfer, 2014; Goyal et al., 2017; Fraccaro et al., 2016). The variational RNN (*VRNN*, (Chung et al., 2015)) is one renowned instantiation in this domain that, assuming specific dependency structures in the generative and inference parts, arrives at the following lower bound on $\log p_\theta(\boldsymbol{x}_{\leq T})$:

$$\mathbb{E}_{q_\phi(\boldsymbol{z}_{\leq T}|\boldsymbol{x}_{\leq T})} \left[ \sum_{t=1}^T \log p_\theta(\boldsymbol{x}_t|\boldsymbol{z}_{\leq t}, \boldsymbol{x}_{<t}) - \mathcal{KL}[q_\phi(\boldsymbol{z}_t|\boldsymbol{x}_{\leq t}, \boldsymbol{z}_{<t}) \parallel p_\theta(\boldsymbol{z}_t|\boldsymbol{x}_{<t}, \boldsymbol{z}_{<t})] \right], \quad (1)$$

where information $\boldsymbol{x}_{<t}, \boldsymbol{z}_{<t}$ is captured via a recurrent neural network $\boldsymbol{h}_t = f_{RNN}(\boldsymbol{x}_t, \boldsymbol{z}_t, \boldsymbol{h}_{t-1})$. Given the temporal and multimodal notion of human movement, sequential generative models constitute a good starting point for designing a framework tailored to multiagent trajectories. However, such approaches only account for the temporal aspect of the problem, but neglect potential social dependencies at each timestep.

**Adding Graph Structure**   As a remedy, sequential data can be augmented by a social dimension $\boldsymbol{x}_{\leq T} = \{\boldsymbol{x}_{\leq T}^{(a)}, \forall a \in \mathcal{A}\}$, where $\boldsymbol{x}_t^{(a)} \in \mathbb{R}^d$ denotes a $d$-dimensional feature representation of agent $a \in \mathcal{A}$ at time $t$ (e.g., its 2D position). Permutation invariant models are a prerequisite for processing sequential sets with potentially divergent cardinality, so a direct adoption of Eq. 1 in multiagent settings would implicitly impose the assumption of social independence across agent trajectories. This assumption is trivially inappropriate for interactive systems; thus, related work proposes sensitive solutions - usually in the form of graph encoding strategies - to capture agent-agent interactions.

Yeh et al. (2019) introduce the graph VRNN (*GVRNN*), which operates within the VRNN framework with graph neural networks (GNNs, Battaglia et al. (2018)) representing agents and their interactions as nodes and edges, respectively. More formally, the architecture for computing the components in Eq. 1 amounts to the following structure:

$$p_\theta(\boldsymbol{z}_t|\boldsymbol{x}_{<t}, \boldsymbol{z}_{<t}) = \mathcal{N}(\boldsymbol{z}_t; \text{GNN}_{prior}(\boldsymbol{h}_{t-1})) \quad (2)$$

$$q_\phi(\boldsymbol{z}_t|\boldsymbol{x}_{\leq t}, \boldsymbol{z}_{<t}) = \mathcal{N}(\boldsymbol{z}_t; \text{GNN}_{enc}([\boldsymbol{x}_t, \boldsymbol{h}_{t-1}])) \quad (3)$$

$$p_\theta(\boldsymbol{x}_t|\boldsymbol{x}_{<t}, \boldsymbol{z}_{\leq t}) = \mathcal{N}(\boldsymbol{x}_t; \text{GNN}_{dec}([\boldsymbol{z}_t, \boldsymbol{h}_{t-1}])), \quad (4)$$

where $\boldsymbol{h}_t$ is the set of recurrent agent states $\boldsymbol{h}_t^{(a)} = f_{RNN}(\boldsymbol{x}_t^{(a)}, \boldsymbol{z}_t^{(a)}, \boldsymbol{h}_{t-1}^{(a)})$. We emphasize that, although factorized, the latent space is not marginally independent across agents since each $\boldsymbol{z}_t^{(a)}$ is conditioned on information of all other entities via the (assumed) fully-connected graph. However, in many spatiotemporal patterns, most of the observed elements are irrelevant or even distracting, and the specific composition of relevant factors can change rapidly. Thus, a better strategy is to explicitly detect semantic classes that describe the underlying structural component before aggregating social information into entangled variables.

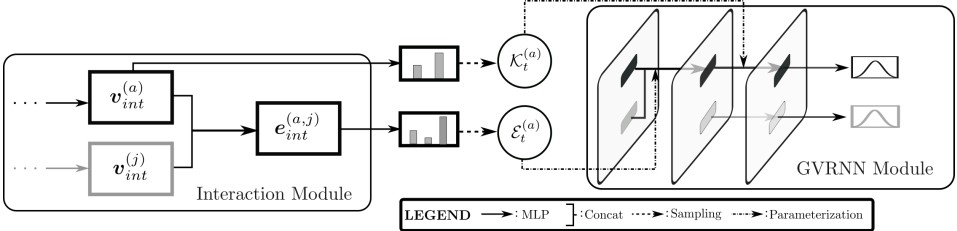

Figure 1: *Left:* Illustration of the output layer of an interaction module (encoder $q_{\phi_1}$ or prior $p_{\theta_2}$) for a target agent $a \in \mathcal{A}$ and an neighboring agent $j \in \mathcal{A}$, where $\boldsymbol{v}$ and $\boldsymbol{e}$ are node and edge embeddings, respectively. *Right:* Illustration of an interaction network (decoder $p_{\theta_1}$, prior $p_{\theta_2}$, or encoder $q_{\phi_2}$) that is parameterized by realized elements from the discrete latent subspace: $\mathcal{E}_t^{(a)}$ picks from multiple node-to-edge MLPs, and $\mathcal{K}_t^{(a)}$ picks from multiple edge-to-node MLPs run in parallel. Thus, implicit knowledge about fundamentally different interaction patterns is encapsulated in the parameters of the edge MLPs associated with the corresponding class. The encoded neighborhood communication $\boldsymbol{e}^{(a,j)}$ contingent on the predicted interaction type, has different behavioral consequences based on the specific role the target agent is currently adopting.

## 3 METHOD

In the following, we propose a novel line of thought on how to model the joint distribution of interactive sequential data $p_\theta(\boldsymbol{x}_{\leq T})$ using variational methods. To address shortcomings of existing methods, our formalization defines disentangled factors of variation that explicitly include a discrete latent subspace describing causal factors that arise from social perspectives. Having conceptualized our modeling goal, we instantiate a concrete architecture termed DIA (Detecting Important Agents) that computes the inferred training components via imposing an inductive bias on the latent space. Appendix B.2 details how the model is trained in an end-to-end fashion.

### 3.1 DERIVING THE TRAINING OBJECTIVE

**Latent Variables**   The interactive systems contemplated in this work are caused by implicit and dynamic structural dependencies that vary both in the type of interaction as well as in the specific role of the interacting participants. For example, in a turnover sequence in soccer, both the role performed (e.g., attacker to defender) and the influential nature (e.g., on the ball agent from ball carrying to pressure exerting) of each player change within a few moments. We propose to explicitly account for such structures via introducing latent causal graphs $\mathcal{G}_t = \{\mathcal{K}_t, \mathcal{E}_t\}$, where $\mathcal{K}_t$ represents the set of discrete agent roles at $t$ (e.g., ball, attacker or defender) $\{\mathcal{K}_t^{(a)}, a \in \mathcal{A}\}$[1], and $\mathcal{E}_t$ encodes their (discrete) pairwise strategies in the form of interaction types $\{\mathcal{E}_t^{(a,j)}, (a,j) \in \mathcal{A} \times \mathcal{A}\}$ (e.g., ball handler, intended pass receiver, etc.). To enable the model to exclude potential social confounders, we externally define an interaction type in $\mathcal{E}_t$ as class "no interaction" while the remaining categories are learned from the data.

However, the structural state captured in $\mathcal{G}_t$ is not sufficiently informative to precisely account for factors that arise from both individual and social perspectives. Therefore, to improve the generative capacity of our model, we inheret latent concepts $\boldsymbol{z}_t$ that account for *all* remaining sources of uncertainty not represented by the inferred causal graphs, yet may vary as a function of $\mathcal{G}_t$.

**Generative Process**   We formally express the generative process by incorporating the above description of causal factors into the definition of our modeling goal, i.e., the marginal likelihood $p_\theta(\boldsymbol{x}_{\leq T})$:

$$p_\theta(\boldsymbol{x}_{\leq T}) = \int_{\mathcal{G}_{\leq T}} \int_{\boldsymbol{z}_{\leq T}} p_\theta(\boldsymbol{x}_{\leq T}, \boldsymbol{z}_{\leq T}, \mathcal{G}_{\leq T}) d\boldsymbol{z}_{\leq T} d\mathcal{G}_{\leq T}$$

$$= \sum_{G_{\leq T}} \int_{\boldsymbol{z}_{\leq T}} \prod_{t=1}^{T} p_{\theta_1}(\boldsymbol{x}_t | \boldsymbol{x}_{<t}, \boldsymbol{z}_{\leq t}, \mathcal{G}_{\leq t}) p_{\theta_2}(\boldsymbol{z}_t | \boldsymbol{x}_{<t}, \boldsymbol{z}_{<t}, \mathcal{G}_{\leq t}) p_{\theta_3}(\mathcal{G}_t | \boldsymbol{x}_{<t}, \boldsymbol{z}_{<t}, \mathcal{G}_{<t}) d\boldsymbol{z}_{\leq T},$$

---

[1]For only a single agent type, we have $\mathcal{K}_t = \mathcal{A}$.

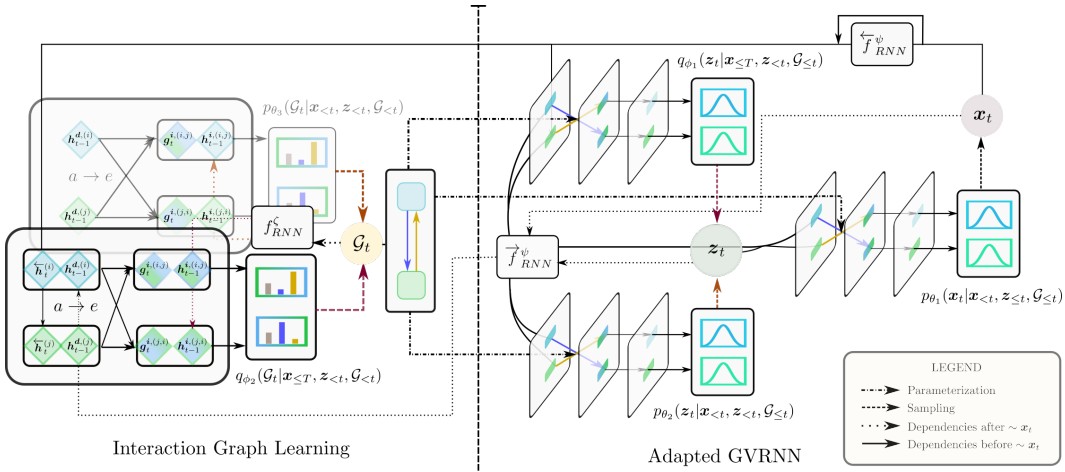

Figure 2: Illustration of the computational dependencies at a timestep for a system with two interacting agents when using one agent type, i.e., $\mathcal{K}_t = \mathcal{A}$. We only show the first layer of the overall two-layer graph network in the interaction part ( left side) to clarify the injection of the GRU states. Red connections denote the parts utilized only during training, orange connections used during inference. We use a diamond shape to indicate the value is deterministic, and use a circle to indicate the variable is stochastic.

where the model likelihood $p_{\theta_1}$ is parameterized using a *decoder*, and $p_{\theta_2}$ and $p_{\theta_3}$ are the *prior* distributions over the specified latent factors.

**Inference Model**  Given the generative model, learning representations of data can be seen as learning a variational approximation of the posterior using an *encoder* that constructs the distribution of latent values for a given observation $\boldsymbol{x}_{\leq T}$. We propose the following factorization:

$$q_\phi(\boldsymbol{z}_{\leq T}, \mathcal{G}_{\leq T} | \boldsymbol{x}_{\leq T}) = \prod_{t=1}^{T} q_{\phi_1}(\mathcal{G}_t | \boldsymbol{x}_{\leq T}, \boldsymbol{z}_{<t}, \mathcal{G}_{<t}) q_{\phi_2}(\boldsymbol{z}_t | \boldsymbol{x}_{\leq T}, \boldsymbol{z}_{<t}, \mathcal{G}_{\leq t}).$$

Note that, as opposed to the VRNN objective, we do impose restrictions on this approximation when defining its dependency structure by including also future inputs $\boldsymbol{x}_{t+1:T}$ in the conditions of $q_{\phi_1}$ and $q_{\phi_2}$. This has been shown to be empirically beneficial in sequential settings (Goyal et al., 2017; Fraccaro et al., 2016).

**Objective Function**  We construct an objective function $\mathcal{L}_{DIA}(\boldsymbol{x}_{\leq T}; \phi, \theta)$ for training the introduced concepts $p_\theta$ and $q_\phi$, by defining a lower-bound on log-likelihood $\log p_\theta(\boldsymbol{x}_{\leq T})$ :

$$\mathbb{E}_{q_\phi(\boldsymbol{z}_{\leq T}, \mathcal{G}_{\leq T} | \boldsymbol{x}_{\leq T})} \left[ \sum_{t=1}^{T} \log \frac{p_{\theta_1}(\boldsymbol{x}_t | \boldsymbol{x}_{<t}, \boldsymbol{z}_{\leq t}, \mathcal{G}_{\leq t}) p_{\theta_2}(\boldsymbol{z}_t | \boldsymbol{x}_{<t}, \boldsymbol{z}_{<t}, \mathcal{G}_{\leq t}) p_{\theta_3}(\mathcal{G}_t | \boldsymbol{x}_{<t}, \boldsymbol{z}_{<t}, \mathcal{G}_{<t})}{q_{\phi_1}(\mathcal{G}_t | \boldsymbol{x}_{\leq T}, \boldsymbol{z}_{<t}, \mathcal{G}_{<t}) q_{\phi_2}(\boldsymbol{z}_t | \boldsymbol{x}_{\leq T}, \boldsymbol{z}_{<t}, \mathcal{G}_{\leq t})} \right],$$
(5)

with $q_{\phi_1}(\mathcal{G}_t | \boldsymbol{x}_{\leq T}, \boldsymbol{z}_{<t}, \mathcal{G}_{<t}) = q_{\phi_1}(\mathcal{K}_t | \boldsymbol{x}_{\leq T}, \boldsymbol{z}_{<t}, \mathcal{G}_{<t}) q_{\phi_1}(\mathcal{E}_t | \boldsymbol{x}_{\leq T}, \boldsymbol{z}_{<t}, \mathcal{G}_{<t})$ and $p_{\theta_3}(\mathcal{G}_t | \boldsymbol{x}_{<t}, \boldsymbol{z}_{<t}, \mathcal{G}_{<t}) = p_{\theta_3}(\mathcal{K}_t | \boldsymbol{x}_{<t}, \boldsymbol{z}_{<t}, \mathcal{G}_{<t}) p_{\theta_3}(\overline{\mathcal{E}}_t | \boldsymbol{x}_{<T}, \boldsymbol{z}_{<t}, \mathcal{G}_{<t})$ . See Appendix A for a derivation of Eq. 5 and an elaboration of the relationship to Eq. 1.

## 3.2 ARCHITECTURAL COMPONENTS

Given past realizations $\boldsymbol{x}_{<t}, \boldsymbol{z}_{<t}, \mathcal{G}_{<t}$, the introduced dependency structure in Eq. 5 requires to first infer the structural component $\mathcal{G}_t$ before subsequently generating variables $\boldsymbol{z}_t$ and $\boldsymbol{x}_t$. Hence, we divide the training procedure at each timestep into two distinct phases, with the first phase addressing the calculation of components over $\mathcal{G}_t$ ($p_{\theta_3}$ and $q_{\phi_1}$), while the subsequent second phase infers components over $\boldsymbol{z}_t$ and $\boldsymbol{x}_t$ ($p_{\theta_1}$, $p_{\theta_2}$ and $q_{\phi_2}$). The overall architecture is illustrated in Figure 2. We describe the computations in more detail below.

**Interaction Graph Learning**  First, we uncover the (dynamic) latent graph space by constructing the encoder (and prior) modules estimating $\mathcal{G}_t$, $\forall t \in \{1, ..., T\}$. Since we define latent concepts $\mathcal{G}_t = \{\mathcal{K}_t, \mathcal{E}_t\}$ in discrete terms (cf. Section 3.1), it is natural to model the encoder (and prior) as a multivariate distribution of independent Categoricals:

$$q_{\phi_1}(\mathcal{K}_t | \boldsymbol{x}_{\leq T}, \boldsymbol{z}_{<t}, \mathcal{G}_{<t}) = \prod_{a \in \mathcal{A}} \text{Cat}\Big(\mathcal{K}_t^{(a)} | f_{enc}^{int}(\boldsymbol{x}_{\leq T}, \boldsymbol{z}_{<t}, \mathcal{G}_{<t}; \phi_1)\Big) \tag{6}$$

$$q_{\phi_1}(\mathcal{E}_t | \boldsymbol{x}_{\leq T}, \boldsymbol{z}_{<t}, \mathcal{G}_{<t}) = \prod_{(i,j) \in \mathcal{A} \times \mathcal{A}} \text{Cat}\Big(\mathcal{E}_t^{(i,j)} | f_{enc}^{int}(\boldsymbol{x}_{\leq T}, \boldsymbol{z}_{<t}, \mathcal{G}_{<t}; \phi_1)\Big), \tag{7}$$

where function $f_{enc}$ is an encoder neural net. Since the interaction space contains the defined "no interaction" label, the derived probability values over $\mathcal{E}_t$ in a Bernoulli context can be interpreted as importance values that encode the level of influence of interacting neighbors on target agents.

While elements in $\mathcal{K}_t$ are anchored to the elements of the agent set, interactive patterns $\mathcal{E}_t$, however, are defined non-symmetrically over all agent pairs. Thus, we impose a function class on $f_{enc}$ that enables learning single-agent as well as pairwise representations based on which we can classify over both node and relation variables. Inspired by Kipf et al. (2018), we employ an adapted variant of GNNs similar to interaction networks (Gilmer et al., 2017) that operates by learning alternating node and edge representations through an iterative process where the learned edge embeddings are used as messages for neighborhood aggregation[2]. See Appendix B.1 for a formalization of the core computations of this GNN variant.

We model the input to $f_{enc}(\boldsymbol{x}_{\leq T}, \boldsymbol{z}_{<t}, \mathcal{G}_{<t})$ via multiple GRU cells (Chung et al., 2014) defined over quantities related to the individual agents. More formally, $f_{enc}$ is given by

$$[\boldsymbol{v}_{enc}^{int}, \boldsymbol{e}_{enc}^{int}] = f_{enc}^{int}(\boldsymbol{x}_{\leq T}, \boldsymbol{z}_{<t}, \mathcal{G}_{<t}) = \text{GNN}_{enc}^{int}([\overrightarrow{\boldsymbol{h}}_{t-1}, \overleftarrow{\boldsymbol{h}}_t], \boldsymbol{h}_{t-1}^{int}; \phi_1), \tag{8}$$

where $[\cdot, \cdot]$ denotes concatenation, $\overrightarrow{\boldsymbol{h}}_{t-1}$ is a set of forward RNN states $f_{RNN}(\boldsymbol{x}_t^{(a)}, \boldsymbol{z}_t^{(a)}, \overrightarrow{\boldsymbol{h}}_{t-1}^{(a)})$, $\overleftarrow{\boldsymbol{h}}_t$ is a set of backward RNN states $f_{RNN}(\boldsymbol{x}_t^{(a)}, \overleftarrow{\boldsymbol{h}}_{t+1}^{(a)})$, $\boldsymbol{h}_{t-1}^{int}$ is a set of hidden states defined along the edges $f_{RNN}(\mathcal{E}_{t-1}^{(a,j)}, \boldsymbol{h}_{t-1}^{(a,j),int})$ of a fully-connected graph structure, and $\boldsymbol{v}_{enc}^{int}, \boldsymbol{e}_{enc}^{int}$ are the node and edge embeddings of the last/output GNN layer, respectively (cf. Figure 1, left). The concatenated agent sets $[\overrightarrow{\boldsymbol{h}}_{t-1}, \overleftarrow{\boldsymbol{h}}_t]$ refer to the initial node representations of the network and the hidden interaction set $\boldsymbol{h}_{t-1}^{int}$ is concatenated with the edge embeddings of the first GNN layer (cf. Figure 2, left). The usage of multiple message passing rounds ($\geq 2$) in conjunction with a fully-connected adjacency matrix allows the model to accomodate structural information from the full input when estimating agent and interaction types.

Finally, we model the distributions comprising Eq. (6) and (7) as $q_{\phi_1}(\mathcal{G}_t | \boldsymbol{x}_{\leq T}, \boldsymbol{z}_{<t}, \mathcal{G}_{<t}) = \text{softmax}(\text{Linear}(f_{enc}^{int}(\boldsymbol{x}_{\leq T}, \boldsymbol{z}_{<t}, \mathcal{G}_{<t}; \phi_1)))$; see Figure 1. The prior distribution $p_{\theta_3}(\mathcal{G}_t | \boldsymbol{x}_{<t}, \boldsymbol{z}_{<t}, \mathcal{G}_{<t})$ is derived analogously to the interaction encoder, omitting the summarized future $\overleftarrow{\boldsymbol{h}}_t$ as GNN input.

**Adapted GVRNN**  Once $\mathcal{G}_t \sim q_{\phi_1}$ is generated, the model needs to reason over the remaining components of Eq. (5). To achieve this, we observe that the functionality of $\mathcal{G}_t$ is grounded in uncovering latent social dependencies at every timestep $t$ within a multiagent segment. Thus, the generated $\mathcal{G}_t$ implicitly characterizes the computational graph at specific junctures and, since predictions are carried out autoregressively, encompasses all information required for computing agent states $\boldsymbol{x}_t$ and $\boldsymbol{z}_t$. Motivated by the previous considerations, we assume $\mathcal{G}_t$ is Markov and thus discard $\mathcal{G}_{<t}$ from the conditions of objects $p_{\theta_1}$, $p_{\theta_2}$ and $q_{\phi_2}$. The remaining computations of Eq. (5) thereby comply with the structure of $\mathcal{G}_t$-conditional VRNN components (cf. Eq. 1 - 4).

In general, we can use any graph encoding strategy performing forecasting on the inferred latent subspace $\mathcal{G}_t$ to model the distributions over $\boldsymbol{x}_t$ and $\boldsymbol{z}_t$. An instantiation with interaction networks constitutes a reasonable design choice in the present context, as there is already empirical evidence

---

[2]Other commonly used graph encoding strategies in the multiagent regime such as graph attention networks (Veličković et al., 2017) or transformers (Vaswani et al., 2017) update node embeddings via learned attention weights and consequently do not explicitly learn vectorial edge representations, rendering them inappropriate for the problem at-hand.

of their effectiveness for modeling sports tracking data (Yeh et al., 2019; Dick et al., 2022). For ease of exposition, we assume an interactive system consisting of two agents and define computations for graph $i \leftarrow j$, where $i \in \mathcal{A}$ and $j \in \mathcal{A}$ as follows.

To accomodate $\mathcal{G}_t$, we parameterize the node and edge functions inherent in each graph network dependent on the predicted classes $\mathcal{E}_t^{(i,j)} \in \{1, ..., K\}$ and $\mathcal{K}_t^{(i)} \in \{1, ..., M\}$. More specifically, we pick corresponding parameters for the edge MLP $f_e$ from the set $\{\psi_1, ..., \psi_K\}$ with $\psi_1 = \emptyset$ denoting the manually defined "no interaction" label and parameters for the node MLP $f_v$ from the set $\{\gamma_1, ..., \gamma_M\}$. The message passing operations for target agent $i$ amount to

$$\boldsymbol{e}^{(i,j)} = f_e([\boldsymbol{v}^{(i)}, \boldsymbol{v}^{(j)}]; \psi_k) \tag{9}$$

$$\boldsymbol{u}^{(i)} = f_v(\boldsymbol{e}^{(i,j)}; \gamma_m). \tag{10}$$

Intuitively, this inductive bias encourages the model to learn semantically meaningful concepts describing distinct movement and interaction patterns and governs attention towards important pieces of data when predicting collective movements. See Figure 1 for a detailed visual depiction of the logic. An overview over the computational dependencies is given in Figure 5b.

## 4 EXPERIMENTS

In this section, we empirically validate our proposed model on two challenging real-world datasets: basketball and soccer data[3] . The dataset details, model architectures, and hyperparameters of both our model and the baseline approaches are discussed in Appendix C.1.

### 4.1 SETUP

**Metrics** We adopt the commonly used Average Displacement Error (ADE) and Final Displacement Error (FDE) metrics to assess the generative performance of future predictions. ADE refers to the $l_2$ error between the predicted locations and the ground truth averaged over the entire trajectory, while FDE is the $l_2$ error for the last predicted point (Alahi et al., 2014). Following prior works, we report the minimum over 20 generated samples.

**Baselines** We compare our method with a variety of SOTA baselines for modeling (i) sports tracking data: *Weak-Sup* (Zhan et al., 2019), *GVRNN* (Yeh et al., 2019), *dNRI* (Graber & Schwing, 2020), *DAG-Net* (Monti et al., 2021), *GRIN* (Li et al., 2021b); and (ii) for modeling urban dynamics: *Joint-$\beta$-cVAE* (Bhattacharyya et al., 2021) and *GRIN* (Li et al., 2021b). To enable direct comparison against this diverse set of methods, we benchmark in two experimental configurations.

Firstly, *DAG-Net* and *Weak-Sup* are inherently advantageous in generating agent trajectories by incorporating future locations in form of heuristically generated labels at prediction time. This renders a direct comparison to fully unsupervised generative methods (like ours) biased. To minimize this bias, we report quantitative results on a rather long prediction horizon of $T_{obs} = 10, T_{pred} = 40$ for both methods. Other baselines such as *GRIN* and *dNRI* cannot be trivially tested on these long-term predictions, so we generate results on configuration $T_{obs} = 40, T_{pred} = 10$. Finally, GVRNN, Joint-$\beta$-cVAE and our framework (DIA) can be applied to the full range of prediction scenarios.

### 4.2 QUANTITATIVE EVALUATION

**Baseline Comparisons** In the first set of experiments, we benchmark our model against recent generative methods for the task of trajectory forecasting. Our empirical findings are summarized in Table 1 and Table 2 (left). DIA emerges as the best generative tool across all tested tasks improving the best (unsupervised) competitor performance by at least 17.3% minADE and 25.5% minFDE for basketball. Remarkably, despite their inherent advantages, DIA also outperforms recent supervised generative methods (Weak-Sup and DAG-Net) by at least 8.3% minADE and 16.8% minFDE. Furthermore, we observe that the methods that learn global latent variables (GRIN and Joint-$\beta$-cVAE) perform significantly worse than the remaining models. Though it may be premature to arrive at a definite conclusion, however the comparatively dismal performance of such baselines in our results

---

[3]The source code will be made publicly available upon acceptance of this manuscript.

|  | Joint-$\beta$-cVAE | GVRNN | dNRI | GRIN | DIA (Ours) |
|---|---|---|---|---|---|
| min ADE | 4.03 | 2.60 | 2.77 | 3.00 | **2.20** |
| min FDE | 6.56 | 5.66 | 5.52 | 6.12 | **4.51** |

|  | Joint-$\beta$-cVAE | GVRNN | Weak-Sup | DAG-Net | DIA (Ours) |
|---|---|---|---|---|---|
| min ADE | 10.64 | 9.73 | 9.47 | 8.98 | **8.29** |
| min FDE | 14.47 | 15.80 | 16.98 | 14.08 | **12.05** |

Table 1: Quantitative results on basketball data (in meters) modeling offensive players for short-term predictions with $T_{pred} = 10$ and $T_{obs} = 40$ (**on top**) and long-term predictions with $T_{pred} = 40$ and $T_{obs} = 10$ (**below**).

| MODEL | MINADE | MINFDE |
|---|---|---|
| JOINT-$\beta$-CVA | 8.10 | 10.91 |
| GVRNN | 7.48 | 10.72 |
| dNRI | 7.60 | 10.88 |
| GRIN | 7.88 | 10.54 |
| DIA (OURS) | **7.02** | **9.68** |

| MODEL | ACCURACY | F1 |
|---|---|---|
| INTERACTION-RNN-DIAG | 0.82 | 0.85 |
| INTERACTION-RNN-FULL | 0.83 | 0.84 |
| INTERACTION-RNN | **0.88** | **0.91** |

Table 2: Results on soccer data. *Left*: Quantitative results for trajectory prediction (in meters) modeling all players with $T_{obs} = 10$ and $T_{pred} = 10$. *Right*: Results for the auxiliary classification task.

tend to suggest limited transferability of such models for sports data. We hypothesize that this is due to lower inherent social interaction patterns in urban environments in relation to the heterogeneous and dynamic causal nature of multiagent systems in the sports domain (Makansi et al., 2022).

**Investigating Latent Structure**   To quantify the benefit of augmenting the latent space with an explicit causal graph $\mathcal{G}_t$, we compare performance metrics of the GVRNN and an adapted DIA version at varying observation and prediction lengths. For this, we modify the DIA architecture such that the second part of the training procedure is identical to the GVRNN computations. In this way, the resulting performance differences capture only influences originating from the proposed graph mechanism. The left part of figure 3 visualizes the resulting numbers. As can be seen, our DIA version yields substantial performance gains, with the performance difference increasing with task complexity, i.e., prediction horizon. We note that an increase in GVRNN complexity is accompanied by a further increase in performance difference, see Appendix C.3. The results thus highlight that modeling relations via attention-based aggregation strategies is insufficient to capture decisive social signals in non-trivial multiagent systems.

The right part of Figure 3 shows predictive results for different agent subgroups when varying the representational capacity of the latent interaction graph $\mathcal{E}_t \in \mathcal{G}_t$. Since separate function approximators are parameterized for each selected interaction class, enlarging the interaction space dimensionality causes overfitting issues, despite higher model expressivity. Thus, the best results are realized by the model configuration with the lowest dimensionality, which still sufficiently encapsulates the underlying system dynamics. Accordingly, this experimental setup can shed light on whether the proposed interaction mechanism $\mathcal{E}_t$ behaves according to our theoretical considerations. For example, in modeling defensive players, we would expect reactive patterns based on the behavior of attacking players, resulting in little intra-group dependencies. Indeed, the results in Figure 3 indicate that our model inherits the expected social behaviors, as the best parameter configuration is in line with the structural complexity of the different player subgroups. Appendix C.4 provides an analogous discussion for agent types $\mathcal{K}_t \in \mathcal{G}_t$.

**Auxiliary Task**   Finally, we examine the extent to which learned latent categories $\mathcal{G}_t$ serve meaningful semantic contributions beyond trajectory generation tasks. To this end, we report quantitative metrics on a classification task using soccer data. In the first step, we assemble a dataset in which the social context at each time step is defined by the estimated agent-ball influences of a DIA training run. That is, agent locations annotated as "no interaction" when considering influences on the ball

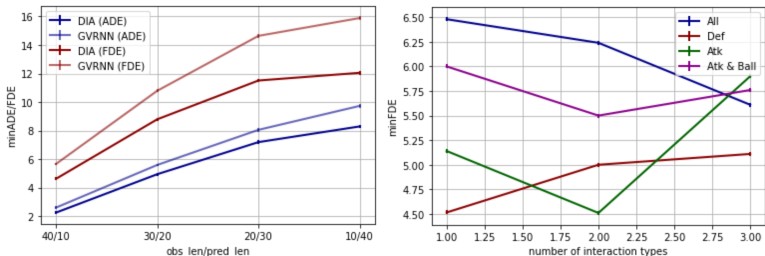

Figure 3: *Left:* Quantitative comparison between GVRNN and an adapted version of our method at different observation and prediction horizons for modeling offensive players. *Right:* Representational capacity of latent $\mathcal{G}_t$ and performance for modeling different agent subgroups. In particular, we fix $\mathcal{K}_t$ and report quantitative results by changing the dimensionality of interaction space $\mathcal{E}_t$.

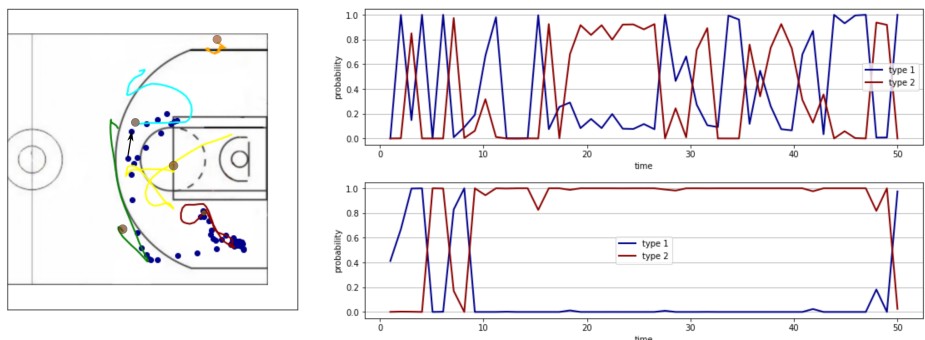

Figure 4: *Left:* Trajectories of the offensive players and the ball (blue dots) from a data point in the test set of the basketball data. *Top right:* Influence $\mathcal{E}^{(b,r)}$ of the red player $r$ on the ball $b$. *Bottom right:* Influence of the ball on the red player.

node are discarded. We then train a spatiotemporal classification network, termed *Interaction-RNN*, for event detection over $\mathcal{Y}$ on the updated data (more details given in Appendix B.3). We compare our method quantitatively against a diagonal (i.e., using ball trajectories only) and a fully-connected (i.e., using all agents for neighborhood aggregation) version of Interaction-RNN in Table 2 (right). The model configuration trained on the data previously extracted by DIA outperforms both baseline configurations. The results indicate potential advantages of injecting interaction structures induced by DIA in downstream applications, as they facilitate detecting key information for learning spatiotemporal classification tasks.

## 4.3    QUALITATIVE EVALUATION

To provide more insights in the learned sequential graphs $\mathcal{G}_{\leq T}$ of DIA, we visually depict a multi-agent segment from the test set and estimated probability values within two interaction categories (and one "no interaction" category) in Figure 4. The upper right image shows the influencing factors of the red player on the ball. This interaction structure is mainly characterized by frequent alternations between the two interaction types with scarce instants of low aggregated influence. Since the interaction values are determined to accurately reflect the input data, the fluctuating pattern observed illustrates highly dynamic social structures in our tested setting.

The figure below encodes the influential structure of the ball as the sending agent on the movements of the red player. Compared to the previous figure, probability values are consistent with one interaction type being active over most of the observed time period. The second interaction type becomes the dominant influential force from the moment the ball is passed to the spatially closest neighbor (green player), and also remains active when the ball reaches the target agent (red player). Thus, the first interaction type encodes more holistic influences, while the second type focuses on more immediate social motion effects. Notably, there are no periods of time in which either of the two interaction modes is inactive. This distinguishes the ball as the central element in basketball and thus

affects the movement of all offensive players at any time. The qualitative results provide further evidence for the ability of our model to learn a semantically meaningful discrete latent subspace. More qualitative results are given in Appendix C.6.

## 5 RELATED WORK

A conceptual line of work adresses human trajectory forecasting as a deterministic regression problem by minimizing the negative log-likelihood and assuming bi-variate Gaussian output distributions Mohamed et al. (2020); Rudenko et al. (2020). While assuming unimodal Gaussians may be sufficient for pedestrian datasets with rather linear trajectories Makansi et al. (2022), they fall short for multimodal trajectory distributions like in team sports Brefeld et al. (2019); so we do not consider these models in this work. Instead, recent methods mostly propose some form of latent variable model, generally formulated as a conditional VAE (Sohn et al., 2015), to capture the stochasticity inherent in future trajectories. At a higher level, the concrete approaches can be broadly categorized by how they model temporal and social dependencies, as well as by their latent space characterizations.

For example, Yeh et al. (2019); Sun et al. (2019); Zhan et al. (2019); Monti et al. (2021) extend the VRNN to handle agent-agent relations via estimating components of Eq. 1 using graph neural networks. Casas et al. (2020b); Salzmann et al. (2020); Bhattacharyya et al. (2021) choose a similar task formulation as well as interaction encoding strategy (Casas et al., 2020a; Anderson et al., 2018), but generate future trajectories non-autoregressively by flattening the time axis into a single dimension. Other conceputally similar approaches (e.g., Girgis et al. (2021); Yuan et al. (2021)) propose transformer-based architectures to encode spatial and temporal relations. There is a plethora of work that aims to improve trajectory forecasting by extending latent information with specific types of agent long-term goals extracted from the trajectory data (Mangalam et al., 2020; 2021; Zhao et al., 2021; Monti et al., 2021; Zhan et al., 2019; Fan et al., 2021; Choi et al., 2021; Girase et al., 2021). However, all these approaches model interactions only implicitly by aggregating messages along the social dimension into spatiotemporal representations. Interestingly, Rudolph et al. (2020) conclude that graph conditional variational methods for predicting multiagent trajectories are often too powerful and simplified methods (e.g., MDNs) should be preferred.

Alternatively, our method can be seen as an novel contribution to the line of research that aims to explicitly infer agent interactions while executing a trajectory prediction task. This task formulation was originally proposed by Kipf et al. (2018), who introduce the NRI framework, a variational autoencoder, where the discrete latent code represents edge predictions in a causal graph. Since the original formulation is limited to learning a single graph for the entire multiagent sequence, several works propose dynamic extensions where the interaction graph can adapt to the conditions per time step (Graber & Schwing, 2020; Li et al., 2020; Gong et al., 2021; Li et al., 2021a). Recently, Löwe et al. (2022) propose to infer causal relations across data points with different underlying interaction graphs but shared dynamics. Other work (Li et al., 2021b) builds upon the NRI framework but leverage a continuous latent space that aims to separate interactive factors from agent intentions.

## 6 CONCLUSION

In this paper, we presented novel framework for modeling the joint distribution of agent trajectories using latent variables. To alleviate shortcomings of previous works, we described the data generation process using disentangled factors of variation that explicitly include a discrete latent subspace reflecting social structures in sports games. We demonstrated that the emerging architecture performs better in predicting trajectories compared to existing strategies and learns informative latent variables. An interesting future line of research is modeling the underlying dynamics at a group level to identify unique team strategies (Raman et al., 2021).

### AUTHOR CONTRIBUTIONS

If you'd like to, you may include a section for author contributions as is done in many journals. This is optional and at the discretion of the authors.

### ACKNOWLEDGMENTS

Use unnumbered third level headings for the acknowledgments. All acknowledgments, including those to funding agencies, go at the end of the paper.

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

# A    ALTERNATIVE PERSPECTIVE ON THE OBJECTIVE FUNCTION

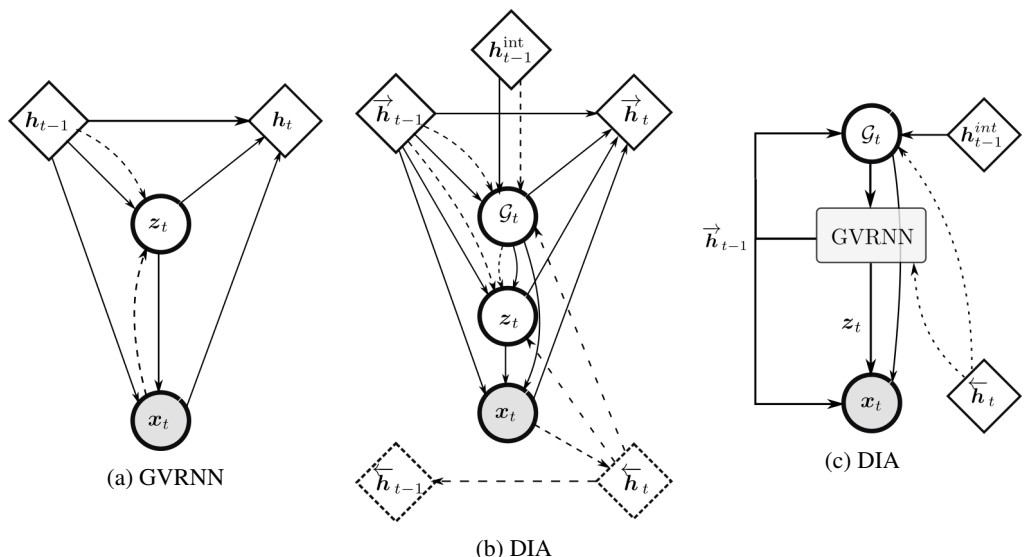

(a) GVRNN

(b) DIA

(c) DIA

Figure 5: Depicting computational dependencies of (a) GVRNN; (b) the proposed interactive sequential latent variable model (DIA); and (c) DIA in relation to GVRNN. Dashed lines indicate the encoding procedures, solid lines the generation process.

Alternatively, Eq. 5 can be written as

$$\sum_{t=1}^{T} \Big( \log p_{\theta_1}(\boldsymbol{x}_t | \boldsymbol{x}_{<t}, \boldsymbol{z}_{\leq t}, \mathcal{G}_{\leq t}) - \mathcal{KL}[q_{\phi_1}(\boldsymbol{z}_t | \boldsymbol{x}_{\leq T}, \boldsymbol{z}_{<t}, \mathcal{G}_{\leq t}) \parallel p_{\theta_2}(\boldsymbol{z}_t | \boldsymbol{x}_{<t}, \boldsymbol{z}_{<t}, \mathcal{G}_{\leq t})]$$

$$- \mathcal{KL}[q_{\phi_2}(\mathcal{G}_t | \boldsymbol{x}_{\leq T}, \boldsymbol{z}_{<t}, \mathcal{G}_{<t}) \parallel p_{\theta_3}(\mathcal{G}_t | \boldsymbol{x}_{<t}, \boldsymbol{z}_{<t}, \mathcal{G}_{<t}) \Big)$$

$$= \sum_{t=1}^{T} \Big( \text{GVRNN}(\overleftarrow{\boldsymbol{h}}_t, \mathcal{G}_t) - \mathcal{KL}[q_{\phi_2}(\mathcal{G}_t(\overleftarrow{\boldsymbol{h}}_t, \overrightarrow{\boldsymbol{h}}_{t-1}, \boldsymbol{h}_{t-1}^{int}) \parallel p_{\theta_3}(\mathcal{G}_t | \overrightarrow{\boldsymbol{h}}_{t-1}, \boldsymbol{h}_{t-1}^{int}) \Big),$$

where $\mathcal{G}_{\leq T}, \boldsymbol{z}_{\leq T} \sim q_\phi(\boldsymbol{z}_{\leq T}, \mathcal{G}_{\leq T} | \boldsymbol{x}_{\leq T})$. See also Figure 5. This can be shown as follows:

$$\int \int q_\phi(\boldsymbol{z}_{\leq T}, \mathcal{G}_{\leq T} | \boldsymbol{x}_{\leq T}) \log \prod_{t=1}^{T} \frac{p_\theta(\boldsymbol{z}_t | \boldsymbol{x}_{<t}, \boldsymbol{z}_{<t}, \mathcal{G}_{\leq t}) p_\theta(\mathcal{G}_t | \boldsymbol{x}_{<t}, \boldsymbol{z}_{<t}, \mathcal{G}_{<t})}{q_\phi(\boldsymbol{z}_t | \boldsymbol{x}_{\leq t}, \boldsymbol{z}_{<t}, \mathcal{G}_{\leq T}) q_\phi(\mathcal{G}_t | \boldsymbol{x}_{\leq t}, \boldsymbol{z}_{\leq t}, \mathcal{G}_{<t})} d\mathcal{G}_{\leq T} d\boldsymbol{z}_{\leq T}$$

$$= \sum_{t=1}^{T} \int \int q_\phi(\boldsymbol{z}_{\leq T}, \mathcal{G}_{\leq T} | \boldsymbol{x}_{\leq T}) \log \frac{p_\theta(\boldsymbol{z}_t | \boldsymbol{x}_{<t}, \boldsymbol{z}_{<t}, \mathcal{G}_{\leq t}) p_\theta(\mathcal{G}_t | \boldsymbol{x}_{<t}, \boldsymbol{z}_{<t}, \mathcal{G}_{<t})}{q_\phi(\boldsymbol{z}_t | \boldsymbol{x}_{\leq t}, \boldsymbol{z}_{<t}, \mathcal{G}_{\leq T}) q_\phi(\mathcal{G}_t | \boldsymbol{x}_{\leq t}, \boldsymbol{z}_{\leq t}, \mathcal{G}_{<t})} d\mathcal{G}_{\leq T} d\boldsymbol{z}_{\leq T}$$

$$= \sum_{t=1}^{T} \int \int q_\phi(\boldsymbol{z}_{\leq t}, \mathcal{G}_{\leq t} | \boldsymbol{x}_{\leq t}) \log \frac{p_\theta(\boldsymbol{z}_t | \boldsymbol{x}_{<t}, \boldsymbol{z}_{<t}, \mathcal{G}_{\leq t}) p_\theta(\mathcal{G}_t | \boldsymbol{x}_{<t}, \boldsymbol{z}_{<t}, \mathcal{G}_{<t})}{q_\phi(\boldsymbol{z}_t | \boldsymbol{x}_{\leq t}, \boldsymbol{z}_{<t}, \mathcal{G}_{\leq t}) q_\phi(\mathcal{G}_t | \boldsymbol{x}_{\leq t}, \boldsymbol{z}_{\leq t}, \mathcal{G}_{<t})} d\boldsymbol{z}_{\leq t} d\mathcal{G}_{\leq t}$$

$$= \sum_{t=1}^{T} \int \int q_\phi(\boldsymbol{z}_{<t}, \mathcal{G}_{<t} | \boldsymbol{x}_{<t}) \bigg( - \mathbb{E}_{q_\phi(\mathcal{G}_t | \boldsymbol{x}_{\leq t}, \boldsymbol{z}_{<t}, \mathcal{G}_{<t})} \big[ \mathcal{KL}[q_\phi(\boldsymbol{z}_t | \boldsymbol{x}_{<t}, \boldsymbol{z}_{<t}, \mathcal{G}_{\leq T} \parallel p_\theta(\boldsymbol{z}_t | \boldsymbol{x}_{<t}, \boldsymbol{z}_{<t}, \mathcal{G}_{<t})]\big]$$

$$- \mathcal{KL}[q_\phi(\mathcal{G}_t | \boldsymbol{x}_{\leq t}, \boldsymbol{z}_{<t}, \mathcal{G}_{<t}) \parallel p_\theta(\mathcal{G}_t | \boldsymbol{x}_{<t}, \boldsymbol{z}_{<t}, \mathcal{G}_{<t})] \bigg) d\boldsymbol{z}_{<t} d\mathcal{G}_{<t}$$

$$= \mathbb{E}_{q_\phi(\boldsymbol{z}_{\leq T}, \mathcal{G}_{\leq T} | \boldsymbol{x}_{\leq T})} \bigg[ \sum_{t=1}^{T} - \mathcal{KL}[q_\phi(\boldsymbol{z}_t | \boldsymbol{x}_{\leq t}, \boldsymbol{z}_{<t}, \mathcal{G}_{\leq T}) \parallel p_\theta(\boldsymbol{z}_t | \boldsymbol{x}_{<t}, \boldsymbol{z}_{<t}, \mathcal{G}_{<t})]$$

$$- \mathcal{KL}[q_\phi(\mathcal{G}_t | \boldsymbol{x}_{\leq t}, \boldsymbol{z}_{<t}, \mathcal{G}_{\leq t}) \parallel p_\theta(\mathcal{G}_t | \boldsymbol{x}_{<t}, \boldsymbol{z}_{<t}, \mathcal{G}_{<t})] \bigg].$$

# B    MODEL DETAILS

## B.1    GRAPH NEURAL NETWORKS

The core operations of the GNN modules comprising the DIA can be formally expressed as follows:

$$v \rightarrow e: \quad \boldsymbol{e}^{(i,j)} = f_e([\boldsymbol{v}_i, \boldsymbol{v}_j]), \tag{11}$$

$$e \rightarrow v: \quad \boldsymbol{o}^{(i)} = f_v(\sum_{j \in N(i)} \boldsymbol{e}^{(i,j)}), \tag{12}$$

where $\boldsymbol{v}_i$ is the input feature representation of agent/node $i$, $\boldsymbol{o}_i$ is the respective node embedding after 1 GNN layer, $N(i)$ denotes the set of agents/nodes that interact with target agent $i$, $\boldsymbol{e}^{(i,j)}$ is the edge embedding of the first GNN layer, and $f_e$ and $f_v$ are the MLPs described above. For the GNNs of the interaction encoding phase, we define 2 rounds of message passing and utilize additional linear layers operating on the node and edge embeddings of the second GNN layer to output $\mathcal{K}_t$ and $\mathcal{E}_t$, respectively. For the adapted GVRNN modules, we use 1 layer as described in the main text with additional MLPs computing the mean and variance vectors of the Gaussians.

## B.2    MODEL TRAINING & TESTING

Instead of absolute positions, our model predicts movements $\Delta \hat{\boldsymbol{x}}_t$ at each timestep. Consequently, we estimate agent locations via $\hat{\boldsymbol{x}}_t = \boldsymbol{x}_{t-1} + \Delta \hat{\boldsymbol{x}}_t$. For simplicity, $\boldsymbol{x}_t$ refers to both relative and absolute positions. For training, the model uses the entire $T = T_{obs} + T_{pred}$ timesteps from ground-truth sequences $\boldsymbol{x}_{\leq T}$. To enable gradient flow through stochastic operations, we use reparametrization and gumbel-softmax trick (Maddison et al., 2017; Jang et al., 2017) for sampling from the encoders over the continuous $\boldsymbol{z}_t$ and the discrete $\mathcal{G}_t$ latent subspace, respectively. .

At test time, we divide the trajectories into an observation and prediction period, where the model only observes the first portion of the ground-truth trajectory $\{\boldsymbol{x}_1, ..., \boldsymbol{x}_{T_{obs}}\}$ and predicts the remaining $T_{pred}$ timesteps autoregressively: $\hat{\boldsymbol{x}}_{T_{obs}+i}, i \in \mathbb{Z}$. Latent variables are sampled from the prior distributions.

## B.3    CLASSIFICATION BASELINE

The proposed classification framework for the spatiotemporal multi-agent regime (*Interaction-RNN*) is essentially geared towards the functionality of an RNN for capturing temporal dependencies of player trajectories. Additionally, to encode interactive agent patterns, we update the inferred RNN states at each time step using an attention-based graph neural network architecture (Veličković et al., 2017). Thus, the updated feature vectors jointly encode social and temporal information. A final softmax layer is then used for classification and the model is optimized by minimizing the cross-entropy loss. Because the labels denote ball-centric events, we use the output of the ball node for loss computation and evaluation.

# C    EXPERIMENTS DETAILS

## C.1    DATA

The *soccer data* contains trajectories of soccer players and the ball extracted from 16 professional soccer matches sampled at 25 frames per second. We assemble a dataset consisting of game excerpts such that the center frames of the sequences at timestep $t = T//2$ correspond to an on-ball event from the set $\mathcal{Y} = \{\text{pass, other ball action, shot, none}\}$. We choose a sequence length of $2s$ and downsample the data to $10Hz$, i.e., $T = 20$. This extraction process yields a total of roughly 34000 multiagent segments divided into 70% training, 15% validation, and 15% test data. We center and normalize the trajectories onto the range $[-1, 1]$ and transform them so that the team in possession of the ball always plays from left to right.

The STATS SportVU *basketball data* consists of tracking data recorded from the 2016 NBA regular season. Every game sequence has two-dimensional positions of 10 players and the ball sampled at 5 frames per second. The data is split into 60% training, 20% validation, and 20% test sets. All data is translated so that the origin of the underlying coordinate system is mapped onto the top-left corner.

### C.2 IMPLEMENTATION DETAILS

**DIA**    All model variants are implemented using PyTorch (Paszke et al., 2019). Training is carried out using Adam (Kingma & Ba, 2014) with default parameters and learning rate of 0.001 and teacher forcing. We select the best performing model using minADE on the validation set. The message passing operations within our graph networks $f_e$ and $f_v$ are 2-layer MLPs with batch normalization (Ioffe & Szegedy, 2015), dropout (Srivastava et al., 2014), and ELU activations (Clevert et al., 2015). In addition, we leverage 2-layer MLPs with LeakyReLU (Xu et al., 2015) activations as feature extractors operating on $z_t$, $x_t$, and $\mathcal{G}_t$. All fully-connected layers are initialized using Xavier initialization (Glorot & Bengio, 2010). For recurrence, we use 2-layer GRU networks (Chung et al., 2014).

**Baselines**    We used the code from the official repositories for baselines (Zhan et al., 2019; Graber & Schwing, 2020; Monti et al., 2021; Li et al., 2021b; Bhattacharyya et al., 2021) and chose the best perfoming hyperparameters from their experiments. For (Yeh et al., 2019), we re-implemented the model by faithfully following the descriptions in their paper. The overall architecture of GVRNN was initially designed such that it is comparable in parameter number to DIA. However, we found that reducing model expressivity led to improved results. If not explicitly stated, the reported numbers correspond to a GVRNN version with approx 84% parameters compared to DIA.

### C.3 DATA & MODEL EFFICIENCY

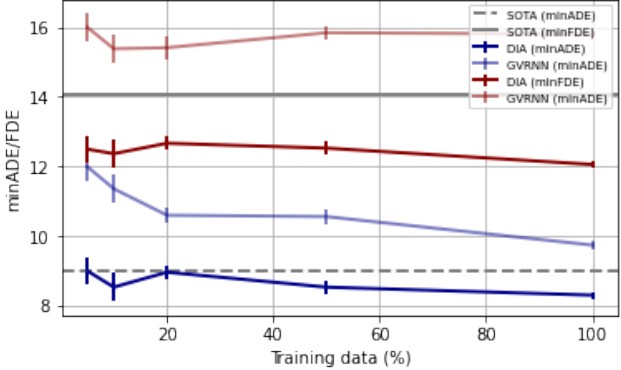

Figure 6: DIA versus GVRNN model performance on long-term prediction for varying data fractions of basketball data. The same data was used across the models. The vertical lines show error bars; the horizontal lines the previously best reported results for this task.

|  | minADE | minFDE | # Parameters |
|---|---|---|---|
| GVRNN (best) | 9.73 | 15.80 | 355588 |
| GVRNN (x2 width) | 9.91 | 16.12 | 733700 |
| GVRNN (x3 width) | 9.94 | 16.20 | 1647364 |
| DIA (no structure) | 10.27 | 15.17 | 400836 |
| DIA | 8.29 | 12.05 | 421768 |

Table 3: Importance of structure versus latent capacity.

To demonstrate the importance of our proposed structure, we perform two sets of additional experiments. First, we directly compare the importance of structure versus larger latent capacity by making adjustments to the GVRNN baseline and our proposed model. To achieve this, we (i) enlarge the encoder/prior and decoder by increasing latent, hidden and rnn dimensions of the GVRNN modules and (ii) remove the proposed graph mechanism from our (adjusted) model so that it essentially resembles a deeper GVRNN. Secondly, we compare generative capabilities of DIA and GVRNN at different data percentages of 5%, 10%, 20%, and 50% of the training data. The training data were randomly selected and the same data is used to train the different models. Hence, the purpose

here is to investigate whether the integrated structure translates into data efficiency by, for example, governing attention to important pieces of data.

Table 3 shows results to investigate the effect of model size on generative capacities. Clearly, larger decoders/encoder/priors yield degenerate results compared to the initial (best) GVRNN configuration. Furthermore, the DIA comparisons highlight the significant importance of enforcing a graph structure since the only difference between both model configurations is our interaction learning strategy. Figure 6 shows the data learning efficiency experiments results with mean and error. Impressively, our model still significantly outperforms the full GVRNN baseline on both error metrics using only 5% (!) of the data. Our model discovers the underlying patterns much faster than GVRNN, whose min ADE numbers improve significantly between 5% and 100% data partitions. Although the results may not be conclusive, they do provide further evidence that our realized improvements do not originate from increased model complexity, but rather because $\mathcal{G}_t$ accurately captures the latent ground-truth factors that arise from social perspectives.

| | # Parameters | Sampling Speed |
|---|---|---|
| DAG-Net | 184040 | 1.4581 |
| GVRNN | 355588 | 0.0764 |
| DIA (Ours) | 421768 | 0.0931 |

Table 4: Parameter size and sampling speed for SOTA models. Models were benchmarked on a Nvidia V100 GPU. The sampling speed is the average over mini-batch long-term predictions ($T_{pred} = 40$) of offensive players.

Table 4 shows parameter sizes and sampling speeds of SOTA methods expressed as averaging over mini-batches for modeling offensive players in basketball. While our model is slower than GVRNN, we emphasize that DIA can reduce runtime by at least 10x compared to GVRNN while maintaining superior results (cf. Figure 6).

## C.4 AGENT TYPES

| # Agent Types | minADE | minFDE |
|---|---|---|
| 1 | 9.18 | 12.87 |
| 2 | 9.06 | 12.74 |
| 3 | 8.94 | 11.64 |
| 4 | 9.00 | 12.71 |

Table 5: Quantitative results on basketball data (in meters) modeling all players with $T_{pred} = 40$ and $T_{obs} = 10$ for varying dimensionality of $\mathcal{K}_t$ and a dimensionality of 1 for the interaction space $\mathcal{E}_t$.

To provide further evidence for potential semantic interpretations of the discrete latent space, we examine the effect of capacity change in agent types $\mathcal{K}_t \in \mathcal{G}_t$ on the generative performance. Table 5 summarizes the results for modeling the dynamics of all players on basketball data. Intuitively, we would expect 3 fundamentally different agent types causing the observed movements since the data consists of attacking and defensive players, and the ball. Similar to Section 4.2, the empirical results confirm this intuition.

## C.5 Continuous Latent Variable

An appropriate local solution in our learning problem accurately describes the data while collecting implicit information into latent space $\{\mathcal{G}_t, \boldsymbol{z}_t\}$. As part of the main paper, we qualitatively and quantitatively investigated the discovered relational structure $\mathcal{G}_t$, finding semantically significant patterns, and demonstrated an accurate approximation of the underlying multimodal data distributions through baseline comparisons. Thus, this section is devoted to quantify the information captured in the continuous subspace $\boldsymbol{z}_t$.

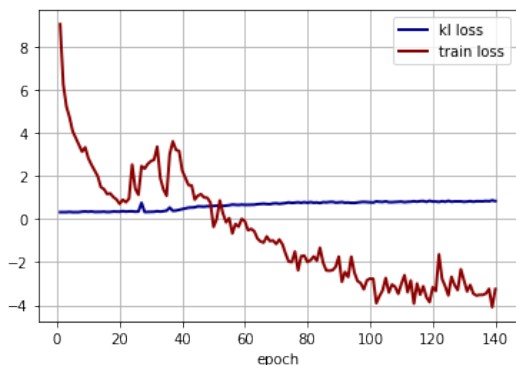

Figure 7: The full training loss $\mathcal{L}_{DIA}$ and KL loss between prior and approximate posterior on $\boldsymbol{z}_{\leq T}$ during a training run on soccer data.

A potential metric to measure the degree of dependency between $\boldsymbol{x}$ and $\boldsymbol{z}$ i.e., to decide whether the continuous latent variable encodes useful information, is by monitoring the KL divergence between the variational posterior and prior. When naively training VAE architectures that consist of a decoder exhibiting sufficiently powerful function approximators with autoregressive dependencies (e.g., RNNs), existing sequential latent variable models frequently report a phenomenon named posterior collapse (Chung et al., 2015), where the model tends to converge to regions of the loss surface that contain bad local minima (or saddle points) at $KL = 0$. Since the latent information the decoder receives is then essentially equivalent to Gaussian noise, the model remits to a standard unconditional RNN, i.e., it learns to generate the input solely based on the autoregressive properties independent of the latent information $\boldsymbol{z}$.

Figure 7 displays (average) values for losses $\mathcal{L}_{DIA}$ and $\mathcal{KL}[q_{\phi_1}(\boldsymbol{z}_t|\boldsymbol{x}_{\leq T}, \boldsymbol{z}_{<t}, \mathcal{G}_{\leq t}) \parallel p_{\theta_2}(\boldsymbol{z}_t|\boldsymbol{x}_{<t}, \boldsymbol{z}_{<t}, \mathcal{G}_{\leq t})]$ during a training run on basketball data. Here we observe a gradual increase in KL loss values that is accompanied by a steady decrease in training loss. Thus, the results indicate an effective exploitation of the continuous subspace $\boldsymbol{z}_t$ to generate future agent movements without the necessity to rely on common optimization strategies such as cost annealing (Bowman et al., 2016; Sønderby et al., 2016).

## C.6 Qualitative Evaluation

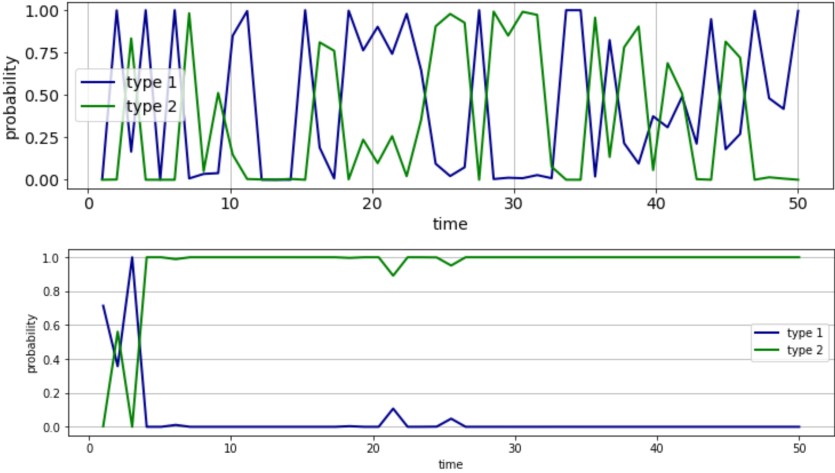

Figure 8: *Top:* Influence of the ball on the green player. *Bottom:* Influence of the green player on the ball.

Figure 8 shows another example of agent-ball interactions for the multi-agent segment from section 4.3. When considering influences on the ball (upper image), we again see a high frequency with respect to the dominant interaction type. When reversing the influence direction, we again observe more constant patterns, where the second interaction type becomes dominant as soon as the ball arrives at the spatially closest agent.

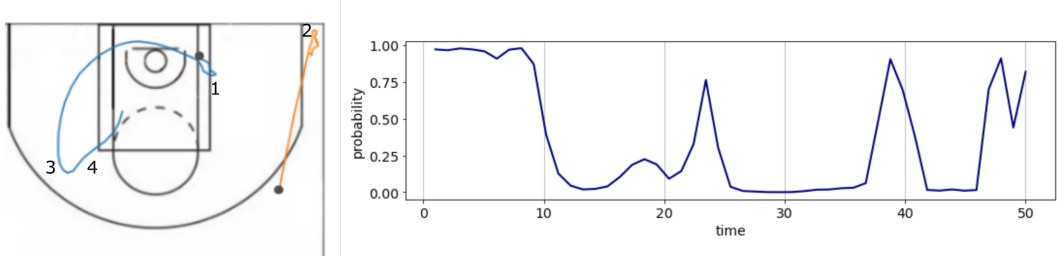

Figure 9: *Left:* Trajectories of two agents. The numbers indicate times at which the probability values of the right graph undergo rapid change. *Right:*

Figure 9 depicts an exemplary timeline for player-player interactions in a Bernoulli setting. Again, our model learns an interpretable semantic structure, which we describe below. The sharp drop in probability values at around $t = 10$ (right plot) is accompanied with the direction change of the blue player (indicated by symbol 1) in the left plot. The probability spike between $t = 20$ and $t = 30$ is caused by the change of direction of the orange player (symbol 2) at the right corner on the pitch. The spike at around $t = 40$ is caused by the blue player turning towards the orange player (symbol 3); the last increase in probability occurs when the blue player accelerates towards the orange player (symbol 4).

