# OpenReview forum: "Interactive Sequential Generative Models"
_ICLR.cc/2023/Conference — Submitted to ICLR 2023_

### Official Review · Reviewer_qCyi · 2022-10-24

**Confidence:** 4
**Correctness:** 3
**Technical Novelty And Significance:** 3
**Empirical Novelty And Significance:** 3
**Recommendation:** 5

**Clarity, Quality, Novelty And Reproducibility:**

The paper does not read well in my opinion, for two reasons in particular: First, several statements are somewhat open-ended and confusing to me such as: “The architecture draws on insights from neural interaction learning to allocate all latent information into the intended variables”, “via attaching latent variables to individual agents, their respective representative capacity is governed by the cardinality of the agent set. Hence, the complexity of information inherent in larger multi-agent systems can be accounted for in a natural way” – making the claims more specific and substantial would greatly help this. Second, important details of the implementation (which clarifies a lot of the actual contribution) is move to the appendix (B1 and B2 should be in the main body in my opinion, which also contain under-explained statements such as “using samples from the encoder and ground-truth” – need to be more specific about “using” – and similarly when explaining the inference time).

The quality is lacking in terms of not providing error bars for experiments.

The work seems somewhat novel to me, as I have not seen this particular structure placed on the latent graph, and setting up this structure and making it train successfully does have merit. But I’m not convinced that the structure is useful, I would like to see some experiments directly comparing the importance of structure vs. larger latent capacity. The setup seems mostly reproducible, but the number of channel dimensions, K, and M seem missing.

**Typos**

Page 2: i index missing in definition of D.

Page 4: it is natural is to

B.1: a vectorial representation

B.2: multiagent sequence

**Strength And Weaknesses:**

**Strengths**

The paper aims to solve an interesting problem with clear applications, and sets up the problem well. The design choices are sufficiently explained (though an ablation study would be nice to verify some of the intuitions).

**Weaknesses**

I have the following concerns regarding the validity of the contributions:

1) The method uses the fully-connected graph structure to infer the discrete latents (agents and type interactions), which are then used to infer the graph structure used for time-series prediction at each step. However, if you are going to use the fully-connected graph as a latent to your latent, why not directly use it as the latent as in GVRNN? To elaborate more, the main contribution of this work in my view is to place a structure on the fully-connected latent graph (since existing works like GVRNN already incorporate the fully-connected latent graph), but if you are going to use the fully-connected graph to learn this additional structure, this seems to essentially defeat the purpose: you are using the fully-connected graph as part of your latent anyways!

2) Since the proposed model is larger (more complex) compared to baselines due to having a set of graphs to choose from instead of a single one (more concretely, the functions involved in encoding/decoding use a discrete set of parameters), how can we be sure that any improvement is not merely due to this increase in complexity? In other words, does simply using “larger” decoders/encoder/priors achieve the same results as your proposed model?

3) I don’t see how the results in Tables 1 and 2 support this statement on page 7: “The empirical results suggest that our proposed modeling strategy effectively both eliminates interfering social signals and discriminates between different agent and interaction types so that future agent movements can be captured more accurately.”

4) None of the reported results show error bars, so it is not clear if the average improvements upon baselines are statistically significant.

**Summary Of The Paper:**

This work proposes to use a specific structure in the latent space of conditional VAEs used for multi-agent time-series prediction. In particular, the paper argues that the existing use of graph neural nets to capture social dependence between agents for predicting future states contains a lot of redundancies due to the fully-connected nature of the assumed underlying graph. To that end, a latent structure is proposed where at each time step first a node/edge labeling of the fully-connected graph of agents is attained, and then the labels are used to select the parameters of the decoder/prior/encoder that are conditioned on the graph, thereby selecting a particular choice among a discrete set of latent graph structures when computing the final state predictions at each time step. The experiments show improvements in time-series prediction of two team sports (soccer and basketball dynamics) over existing methods, as well as some evidence for learning a sensible latent graph structure.

**Summary Of The Review:**

The paper targets an interesting problem: placing a structure on the latent graph for multi-agent time-series prediction; but it lacks clarity in its writing, and its experiments are not convincing as detailed above.

---
### Post-rebuttal update
I thank the authors for addressing my concerns. I've raised my score to 5 since the paper now reads better and includes experiments clarifying the importance of the added latent structure. However, since results still lack error bars (all except the new figure), and also it is not explained how the error bars are obtained, I cannot be sure of the significance of the contributions. Nonetheless, I think the paper's main idea is valuable and interesting, and I encourage the authors to rewrite the paper with more focus on statistically robust and well-explained comparisons with state-of-the-art methods.

---

> ### Author Response · Authors · 2022-11-18
> **Response to Reviewer qCyi 1**
>
> We would like to thank you for your praise and constructive suggestions for improving the paper. We believe that some of the key concerns you have raised actually address the same underlying point, which we hope to mitigate with the following clarifications and additional experiments. We first present our empirical arguments and then give potential conceptual explanations.
>
> *Since the proposed model is larger (more complex) compared to baselines due to having a set of graphs to choose from instead of a single one (more concretely, the functions involved in encoding/decoding use a discrete set of parameters), how can we be sure that any improvement is not merely due to this increase in complexity? In other words, does simply using “larger” decoders/encoder/priors achieve the same results as your proposed model?*
>
> *But I’m not convinced that the structure is useful, I would like to see some experiments directly comparing the importance of structure vs. larger latent capacity.*
>
> The question whether the achieved generative performance improvements are based solely on increased model complexity is indeed a very relevant question. We begin by summarizing additional experiments conducted that to address this concern. Subsequently, we elaborate on empirical evidence that was already included in the original version of the paper and clarifying edits we made to emphasize this.
>
> *Additional experiments:**
>
> To demonstrate the importance of our proposed structure, we perform two sets of additional experiments.
>
> 1. We directly compare the importance of structure vs. larger latent capacity by making adjustments to the GVRNN baseline and our proposed model. In particular, we experiment along two dimensions:
>     1. We enlarge the encoder/prior and decoder by increasing latent, hidden and rnn dimensions of the GVRNN modules.
>     2. We remove the proposed graph mechanism from our (adjusted) model so that it essentially resembles a deeper GVRNN.
> 2. We compare generative capabilities of DIA and GVRNN at different data percentages of 5%, 10%, 20%, and 50% of the training data. The training data were randomly selected and the same data is used to train the different models.  Hence, the purpose here is to investigate whether the integrated structure translates into data efficiency by, for example, governing attention to important pieces of data.
>
> Our results are as follows.
>
> 1. The table below shows results to investigate the effect of model size on generative capacities. Clearly, larger decoders/encoder/priors yield degenerate results compared to the initial (best) GVRNN configuration. Furthermore, the DIA comparisons highlight the significant importance of enforcing a graph structure since the only difference between both model configurations is our interaction learning strategy.
>
>
>     |  | minADE | minFDE | # Parameters |
>     | --- | --- | --- | --- |
>     | GVRNN (best) | 9.73 | 15.80 | 355588 |
>     | GVRNN (x2 width) | 9.91 | 16.12 | 733700 |
>     | GVRNN (x3 width) | 9.94 | 16.20 | 1647364 |
>     | DIA (no structure) | 10.27 | 15.17 | 400836 |
>     | DIA | 8.29 | 12.05 | 421768 |
>
> 2. The figure below shows the data learning efficiency experiments results with mean and error. Impressively, our model still *significantly* outperforms the full **GVRNN baseline on both error metrics using only 5% (!) of the data. Our model discovers the underlying patterns much faster than GVRNN, whose min ADE numbers improve significantly between 5% and 100% data partitions. Although the results may not be conclusive, they do provide further evidence that our realized improvements do not originate from increased model complexity, but rather because G_t accurately captures the latent ground-truth factors that arise from social perspectives.
>
> Note: The Figure can be also accessed through the following anonymous link (and Appendix C.3 in revised manuscript): https://ibb.co/cc5HM5y
>
> (cont.)

---

> > ### Author Response · Authors · 2022-11-18
> > **Response to Reviewer qCyi 2**
> >
> > **Evidence from the original submission:**
> >
> > Although not explicitly mentioned in our submitted version, we have indeed already experimented with larger GVRNN capacities in prelimenary experiments (due to the exact concern you raised), thereby arriving at the aforementioned observations.  We agree that this is important to describe in more detail, so we have included a section in the appendix that discusses the model architectures and hyperparameters of our model as well as the baseline models. From this perspective, the importance of our alorithmic contribution was already apparent in the original submission. We summarize below.
> >
> > - Figure 3 (left) provides GVRNN comparisons with a modified DIA version for different observation and prediction periods. The results obtained there show that "modeling relationships via attention-based aggregation strategies is insufficient to capture decisive social signals in non-trivial multiagent systems," highlighting the importance of introducing explicit social interaction learning.
> > - Figure 3 (right) and Appendix C.4 show declining performances when social representation capacity exceeds expected social complexity. This indicates that enlarging model expressivity does not correspond automatically better results; at least in the context of our model.
> > - Qualitative evlautions in Section 4.3 and Appendix C.6 indicate that the model does learn an interpretable sematic structure. Futhermore, in Section 4.2. (paragraph “Auxiliary Task”), we provide quantitative evidence for the model capturing some notion of importance between agents that can be leveraged effectively in downstream applications.
> >
> > *The method uses the fully-connected graph structure to infer the discrete latents (agents and type interactions), which are then used to infer the graph structure used for time-series prediction at each step. However, if you are going to use the fully-connected graph as a latent to your latent, why not directly use it as the latent as in GVRNN? ****To elaborate more, the main contribution of this work in my view is to place a structure on the fully-connected latent graph (since existing works like GVRNN already incorporate the fully-connected latent graph), but if you are going to use the fully-connected graph to learn this additional structure, this seems to essentially defeat the purpose: you are using the fully-connected graph as part of your latent anyways!*
> >
> > To our understanding, the argument is as follows: Since we assume a fully connected graph to derive agent and interaction categories, our model can in principle be structured within the GVRNN framework since G_t is conditioned on its components and inferred via fully-connected graph networks. Given sufficient model capacity (e.g., by adding GNN layers or increasing the latent dimension), such a model would then capture the same interaction patterns internally. Although this argument may seem conceptually sound at first glance, it is precisely this model structure that renders our goals feasible in the first place. The empirical results described above successfully confirm this. In addition, we would like to outline potential conceptual reasons for the comparatively poor results:
> > 1. Pairwise relationships among heterogenous agents are a significant source of uncertainty in our contemplated multiagent systems that are both dynamic and infeasible to annotate manually. To address this inherent multimodality, we need to account for such factors as part of a latent space.  Although it is theoretically possible to capture all latent factors into only single latent variables (as done in, e.g., in GVRNN), there is ample evidence from the VAE literature that defining a disentangled latent space in conjunction with inductive biases that encourage the defined disentanglement (e.g., some form of supervision or constraints) produces significant performance improvements with relation to a single latent space (see, e.g., [A,B,C,D]). Thus, we provide i) conceptual contributions on how we should think about variational methods for interactive sequential data and (ii) a more algorithmically oriented realization of this new line of thought including an inductive bias for structuring the latent space. Our empirical results clearly show the benefits of having a partitioned latent space.
> >
> > (cont.)

---

> > > ### Author Response · Authors · 2022-11-18
> > > **Response to Reviewer qCyi 3**
> > >
> > > 2. Attention-based aggregation mechanisms in graph networks tend to assign non-zero attention weights to irrelevant or unimportant elements, which dilutes the attention given to truly significant information (cf., e.g. [E,F]). However, our tested data consists of potentially many a) irrelvant agents and b) different interaction types. In contrast, we force the model to take binary decisions over potential multiple social categories via (differentiable) sampling from a discrete latent graph. This strategy forces the model to pay attention to the relevant information while discard the others entirely to reduce information redundancy in the subsequent predictions. Importantly, we perform this information compression assuming a fully connected graph, as otherwise we would eliminate potential interactions before seeing any data.
> > > 3. A unified GVRNN framework where conditioning on G_t is simply replaced with internal representations of a deeper graph neural network has architectural drawbacks. Since GVRNN starts already with a fully-connected graph, adding more layers does not add to the receptive field and thus expressive power of the network. In fact, it tends to be harmful (cf., e.g., [G]). We tested this empirically.
> > > 4. Our representation learning approach increases the range of possible applications. Indeed, many of our empirical contributions would not be possible without uncovering explicit social patterns in a latent variable.
> > >
> > > [A] Dai, Andrew M., and Quoc V. Le. "Semi-supervised sequence learning." *Advances in neural information processing systems* 28 (2015).
> > >
> > > [B] Ilse, Maximilian, et al. "Diva: Domain invariant variational autoencoders." *Medical Imaging with Deep Learning*
> > > . PMLR, 2020.
> > >
> > > [C] Joy, Tom, et al. "Capturing Label Characteristics in VAEs." *International Conference on Learning Representations*. 2021.
> > >
> > > [D] Li, Longyuan, et al. "Grin: Generative relation and intention network for multi-agent trajectory prediction." *Advances in Neural Information Processing Systems*  34 (2021): 27107-27118.
> > >
> > > [E] Shen, Tao, et al. "Reinforced self-attention network: a hybrid of hard and soft attention for sequence modeling." *arXiv preprint arXiv:1801.10296* (2018).
> > >
> > > [F] Anirudh Vemula, Katharina Muelling, and Jean Oh. “Social attention: Modeling attention in human crowds.” In *2018 IEEE international Conference on Robotics and Automation (ICRA*), pages 1–7.
> > >
> > > [G] Cai, Chen, and Yusu Wang. "A note on over-smoothing for graph neural networks." *arXiv preprint arXiv:2006.13318* (2020).
> > >
> > >
> > > *I don’t see how the results in Tables 1 and 2 support this statement on page 7: “The empirical results suggest that our proposed modeling strategy effectively both eliminates interfering social signals and discriminates between different agent and interaction types so that future agent movements can be captured more accurately.”*
> > >
> > > Thank you for pointing this out. We agree that the baseline comparisons on their own do not allow for this statement. We have therefore deleted the sentence.
> > >
> > > *The paper does not read well in my opinion, for two reasons in particular: First, several statements are somewhat open-ended and confusing to me such as:*
> > >
> > > - *“The architecture draws on insights from neural interaction learning to allocate all latent information into the intended variables”,*
> > >
> > > Here we refer to the inductive bias introduced to separate social and remaining latent factors of variation by storing them in the variables intended for it (G_t and z_t). We agree that this sentence can be confusing at this point in the manuscript and have updated the introduction to Section 3 accordingly
> > >
> > > - *“via attaching latent variables to individual agents, their respective representative capacity is governed by the cardinality of the agent set. Hence, the complexity of information inherent in larger multi-agent systems can be accounted for in a natural way” – making the claims more specific and substantial would greatly help this*
> > >
> > > This implies that the dimensionality of both latent variables changes as a function of the number of agents since we assign a separate variable to each agent. Thus, adding new agents automatically increases latent capacity without any external adjustments. We have removed the sentence as it is neither relevant to our goals nor distinguishes our paper from existing work.
> > >
> > > In addition, we have made general clarifying edits throughout the paper:
> > >
> > > - We edited the description of the latent variables to express our modeling goals more clearly
> > > - We incorporated a novel baseline suggested by R1 and added a discussion on the advantages/disadvanteges on the transferability of models for urban environments to sports datasets in the baseline comparisons

---

> > > > ### Author Response · Authors · 2022-11-18
> > > > **Response to Reviewer qCyi 4**
> > > >
> > > > *Second, important details of the implementation (which clarifies a lot of the actual contribution) is move to the appendix (B1 and B2 should be in the main body in my opinion, which also contain under-explained statements such as “using samples from the encoder and ground-truth” – need to be more specific about “using” – and similarly when explaining the inference time).*
> > > >
> > > > We agree with this. In particular, Appendix B1 is of high importance since it provides a mathematical description of Figure 1. We integrated B1 into the main text. In addition, we made clarifying edits to Section B2 (now separated into Appendix B.1 and B.2), which we will likewise add to the main text upon acceptance of the manuscript.
> > > >
> > > > *None of the reported results show error bars, so it is not clear if the average improvements upon baselines are statistically significant.*
> > > >
> > > > We added error bars to the respective visualizations, which highlight our quite substantial improvements in relation to recent approaches. However, we note here that reporting average minADE/FDE values is generally sufficient when quantitatively comparing generative models for sequential multiagent data (cf., e.g., [A, B, C, D, E] and others). We hypothesize that this is due to the robustness of minADE/FDE values, since they are already calculated based on minimum values over several samples from latent space.
> > > >
> > > > [A] Salzmann, Tim, et al. "Trajectron++: Dynamically-feasible trajectory forecasting with heterogeneous data." *European Conference on Computer Vision*. Springer, Cham, 2020.
> > > >
> > > > [B] Yuan, Ye, et al. "Agentformer: Agent-aware transformers for socio-temporal multi-agent forecasting." *Proceedings of the IEEE/CVF International Conference on Computer Vision*. 2021.
> > > >
> > > > [C] Monti, Alessio, et al. "Dag-net: Double attentive graph neural network for trajectory forecasting." *2020 25th International Conference on Pattern Recognition (ICPR)*. IEEE, 2021.
> > > >
> > > > [D] Mangalam, Karttikeya, et al. "It is not the journey but the destination: Endpoint conditioned trajectory prediction." *European conference on computer vision*. Springer, Cham, 2020.
> > > >
> > > > [E] Mangalam, Karttikeya, et al. "From goals, waypoints & paths to long term human trajectory forecasting." *Proceedings of the IEEE/CVF International Conference on Computer Vision*. 2021.
> > > >
> > > > *Typos: Page 2: i index missing in definition of D; Page 4: it is natural is to; B.1: a vectorial representation; B.2: multiagent sequence*
> > > >
> > > > Thanks. We corrected the typos.

---

> ### Author Response · Authors · 2022-12-02
> **Thanks for your update**
>
> Thanks for following up.
>
> We note that we updated the other figures with error bars, too (however they are so small that they are barely visible). Since the other models operate within the same error ranges, we have refrained from reporting them in the tables (as is also usually done in related work). We report on averages over five runs; error bars indicate standard error.

---

### Official Review · Reviewer_yCKo · 2022-10-24

**Confidence:** 3
**Correctness:** 3
**Technical Novelty And Significance:** 3
**Empirical Novelty And Significance:** 4
**Recommendation:** 5

**Clarity, Quality, Novelty And Reproducibility:**

The quality was also good in total because of the above contributions and strengths, but I found the above unclear points. The experiments may not be reproduced because they did not provide the code.

**Strength And Weaknesses:**

Strength:
* The first two above contributions may have novelties.
* The experimental results clearly show the superiority of the proposed approach.

Weakness:
* The presentation of the motivation and what the authors did may be clear, but why they did so compared to the existing methods and specific technical contributions may be unclear (e.g., specific formulations and network architectures). Moreover, there may be no ablation study, so the empirical (quantitative) contributions were also unclear.
* For other unclear points, I described the following specific comments.

Specific comments
* Where is the DIA explanation (may be the proposed method)?
* What was the soccer dataset the authors used? I did not find the details.


**Summary Of The Paper:**

The authors proposed a framework for multiagent trajectories that augments sequential generative models with latent social structures. The contributions are as follows:

1. The authors modeled dynamic dependencies among heterogeneous agents as a graph comprising categorical agent roles and pairwise interactions
2. The authors proposed the neural network architecture combined with interaction graph learning and adaptive GVRNN (graph variational RNN) based on GVRNN ( Yeh et al. 2019)
3. They validated their model on data from professional soccer and basketball and show not only improving upon existing state-of-the-art approaches in forecasting trajectories, but also inferring semantically meaningful representations.


**Summary Of The Review:**

Based on the comment above, I consider that the strength of this paper outperformed the weakness, but I had some concerns at this stage, so I did not give a higher rating.

(After seeing other reviews)
After seeing other reviewers' comments, I consider my rating was too high because I almost agree with these opinions. Therefore, I decreased my rating.

---

> ### Author Response · Authors · 2022-11-18
> **Response to Reviewer yCKo**
>
> Thanks for your feedback. We would specifically like to acknowledge your positive words with relation to the contribution to the field and the quality of the experiments. We hope the following response addresses your remaining concerns.
>
> *The presentation of the motivation and what the authors did may be clear, but why they did so compared to the existing methods and specific technical contributions may be unclear (e.g., specific formulations and network architectures).*
>
> The contribution of the paper goes beyond a single algorithmic approach: it provides substantial new insights into how we should conceive of variational methods for interactive sequential data. The interactive systems contemplated in this work are caused by implicit and dynamic structural dependencies among heterogeneous agents that are often stochastic and infeasible (or very difficult) to annotate manually.  We argue that methods that entangle all sources of uncertainty into only single variables via attention-based aggregation mechanisms are insufficient for capturing all latent information of such complex systems and restrict the range of possible applications. Instead, we emphasize the importance of causal factors that arise from social perspectives by suggesting to separate them from the rest. We ****operate within the variational autoencoder family since instantiations within this model class allow us to inherently address all our considerations sequentially via  a) the construction of a generative process with disentangled latent variables; b) the definition of an variational distribution over the introduced latents; and c) the formalization of a learning problem via deriving the marginal lower-bound.
>
> We then proceed by proposing a more algorithmically oriented realization of this new line of thought and how one might wish to go about structuring the latent space. From Eq (9) and Eq (10), the predicted agent and interaction types define the parameters of all subsequent graph networks that infer variables x_t and z_t. Thus, the realized values in G_t affect movement predictions both directly (through the decoder) and indirectly (throuph prior and encoder): it defines the actual information that is collected over the agent dimension for carrying out those predictions. Since we minimize a reconstruction metric, this ensures some notion of social causality in G_t. We emphasize though that we feel that it is a real strength of the paper to have both the important conceptual contributions that or model conveys, while further proposing a concrete architecture of these to allow their effective use in practice.
>
> In general, we have made some clarifying changes that better express the above contributions. For example,
>
> - We incorporated a novel baseline suggested by R1 and added a discussion on the advantages/disadvanteges on the transferability of models for urban environments to sports datasets in the baseline comparisons
> - We edited the description of the latent variables to express our modeling goals more clearly
> - We updated the paragraph “Adapted GVRNN” by using concrete formulas instead of verbal descriptions
>
> *Where is the DIA explanation (may be the proposed method)?*
>
> DIA (**D**etecting **I**mportant **A**gents) is the name of our architecture. We updated the manuscript to emphasize this.
>
> *What was the soccer dataset the authors used? I did not find the details.*
>
> Unfortunately, we cannot release the soccer data but will add a pointer to the company that owns the rights. (However, the used basketball data is publicly available.)

---

> > ### Author Response · Authors · 2022-11-18
> > **Response to Reviewer yCKo 2**
> >
> > *Moreover, there may be no ablation study, so the empirical (quantitative) contributions were also unclear.*
> >
> > The central theme of our algorithmic contribution is to introduce explicit social patterns that encapsulate informative properties for generative tasks as well as downstream applications. In our paper, we therefore performed ablation studies targeting these goals:
> >
> > - In Section 4.2 (paragraph “Investigating Latent Structures”), we first quantified the impact of learned social categories on generative predictions by effectively removing the proposed interaction graph learning strategies. We then assessed the ability to capture expected structural complexity by varying representational capacities for different agent subgroups.
> > - In the subsequent paragraph “Auxiliary Task”, we tested representation learning properties in a downstream classification task via defining the training data for the classifier using the social context inferred by DIA.
> >
> > At R4's request, we conducted additional ablation experiments:
> >
> > - We directly compare the importance of structure vs. larger latent capacity and report the results in the Table below. Clearly, larger decoders/encoder/priors yield degenerate results compared to the initial (best) GVRNN configuration. Furthermore, the DIA comparisons highlight the significant importance of enforcing a graph structure since the only difference between both model configurations is our interaction learning strategy. We added the table and a short discussion to Appendix C3
> >
> > |  | minADE | minFDE | # Parameters |
> > | --- | --- | --- | --- |
> > | GVRNN (best) | 9.73 | 15.80 | 355588 |
> > | GVRNN (x2 width) | 9.81 | 15.94 | 733700 |
> > | GVRNN (x3 width) | 9.94 | 16.20 | 1647364 |
> > | DIA (no structure) | 10.27 | 15.17 | 400836 |
> > | DIA | 8.29 | 12.05 | 421768 |
> > - We compare DIA and GVRNN generative model performance for varying data fractions of 5%, 10%, 20% and 50% of the training data. This set of experiments thus aims at investigating whether the learned structure translates into data efficiency by, for example, governing attention to important pieces of data. Impressively, our model still *significantly* outperforms the full **GVRNN baseline on both error metrics using only 5% (!) of the available training data.
> > **Note:** The Figure can be also accessed through the following anonymous link (and Appendix C.3 in revised manuscript): https://ibb.co/cc5HM5y
> >
> > *The experiments may not be reproduced because they did not provide the code.*
> >
> > We will provide the source code upon acceptance of the manuscript. Furthermore, we added more implementation details to B1 and B2.

---

### Official Review · Reviewer_Gms6 · 2022-10-25

**Confidence:** 3
**Correctness:** 1
**Technical Novelty And Significance:** 1
**Empirical Novelty And Significance:** 3
**Recommendation:** 3

**Clarity, Quality, Novelty And Reproducibility:**

* Clarity: The paper is largely clear in writing. There are some broad overclaims in terms of modeling intention or causality, neither of which are clearly defined in context, which is a crucial problem

* Quality: The quality can significantly be improved through a  principled grounding of the previously mentioned factors in a fairer and more complete review of the relevant literature, some of which I have linked to in the previous section.

* Novelty: The novelty is limited; as mentioned, conditioning on intentions, incorporating variational graph structures, and modeling social interactions between agents has been done for trajectory and general cue forecasting. This does not prevent an important contribution in this space by itself, but the paper does not explicitly provide one by diligently defining and modeling the factors they claim to do.

* Reproducibility: The paper provides implementation details in the Appendix that is useful for reproducibiliy

**Strength And Weaknesses:**

## Strengths

1. The quantitative evaluation is extensive, including baseline comparisons, investigating the latent structure, and an auxiliary task.
2. The alternate perspective on the objective function in the Appendix is insightful

## Weaknesses

### 1\. Literature Review

The paper regrettably fails to acknowledge a vast body of related literature, on (i) intention-conditioned trajectory prediction, (ii) variational graph methods for trajectory prediction, and (iii) models that explicitly model social interactions for forecasting. At the very least, these references ought to be mentioned and discussed for a diligent representation of the research space, even if the methods are not directly compared against.

(i) **Intention-Conditioned Trajectory Prediction**:

[R1, R2, R3] talk about intention-conditioned trajectory prediction for autonomous vehicles. Apart from the data the methods are applied to, the architectures can be applicable to, and are relevant for, the problem being addressed here. Crucially, the DROGON paper defines intention explicitly (more on this in Weakness 2. below).

(ii) **Variational Graph Methods**:

[R4] from the Neurips I Can't Believe It's Not Better Workshop explicitly deals with graph conditional variational methods for multi-agent trajectory prediction. The results in that paper are very relevant for this research area and should be included.

(iii) **Encoding Social Interactions**:

Graph and other stochastic methods that encode social interactions between agents have been long applied to trajectory and behavior forecasating problems. [R5] explicitly incorporates a spatiotemporal graph for incorporating social interactions between agents. [R6] more recently explicitly takes a meta-learning approach for modeling the dynamics unique to a group for probabilistic forecasting. A sports team is a group, and if each team is viewed as having unique social dynamics resulting from the team's strategy then [R6]'s core modeling idea is directly applicable. The cue in [R6] terms is simply player location here. Their modeling of social influence of other agents is also permutation invariant, a limitation this paper claims about existing methods.

#### References:

[R1] DROGON: A Trajectory Prediction Model based on Intention-Conditioned Behavior Reasoning - Choi et al.

[R2] Intention-Driven Trajectory Prediction for Autonomous Driving - Fan et al.

[R3] LOKI: Long Term and Key Intentions for Trajectory Prediction - Girase et al.

[R4] Graph Conditional Variational Models: Too Complex for Multiagent Trajectories? - Rudolph et al.

[R5] Social-STGCNN: A Social Spatio-Temporal Graph Convolutional Neural Network for Human Trajectory Prediction - Mohamed et al.

[R6] Social Processes: Self-Supervised Meta-Learning over Conversational Groups for Forecasting Nonverbal Social Cues - Raman et al.

### 2\. Unsupported claims and definitions

The paper doesn't actually define agent intentions and causality in the specific setting, so there is no reasonable way to evaluate whether the proposed method actually models intentions. The intention-conditioned trajectory works I've mentioned talk about intention over long- and short- time horizons, where e.g. the former is in terms of goal destinations. Here the paper is talking about team sports with player intentions but simply states that this results from communication. What does intention mean here? Also, the paper claims to model causal relationships, but I can't see any explicit causal factors modeled of learned in the graph structure. There might be other exogenous variables explaining trajectory behavior.

### 3\. Notation

There are a few notational errors. For instance, the variable used for the sequence cannot be the same as the individual elements: $x_{<t} = [x_1, ...]$. See [R4] for this. In many places there exist grammatical errors and incomplete sentences. Please do a pass to fix these.

**Summary Of The Paper:**

This paper claims to incorporate latent factors such as agent intents for the task of multiagent trajectory prediction. It claims to do so by explicitly encoding social structures in sports games to enhance graph-structured latent variable models. Specifically, the method models categorical agent roles and pairwise interactions through what the authors claim is a causal graph. The method is evaluated on professional soccer and basketball game data and performs better than the current state of the art.

**Summary Of The Review:**

The paper fails to define and model agent intentions or causality in trajectory forecasting in a principled manner as they claim to do in this work. This, combined with the lack of fair coverage of relevant literature that represents the concepts and modeling approaches dealt with in this paper prevents me from suggesting an accept.

---

> ### Author Response · Authors · 2022-11-18
> **Response to Reviewer Gms6 1**
>
> We would like to acknowledge your positive words regarding the alternative conceptual perspective and the quality of the experiments. We believe that most of the raised concerns actually stem from misunderstandings. So thank you for pointing us to these details, we now updated the text in all related paragraphs to make things clearer based on your comments. Please find detailed responses below.
>
> ### Literature Review
>
> *The paper regrettably fails to acknowledge a vast body of related literature, on (i) intention-conditioned trajectory prediction, (ii) variational graph methods for trajectory prediction, and (iii) models that explicitly model social interactions for forecasting.*
>
> Though we have gladly added all references to the paper, we point out that they are of less relevance for both the multiagent domain as a whole and our work in particular than many other works already discussed and compared to in all the mentioned categories.
>
> **Intention-Conidtion Trajectory Prediction**
>
> *[R1, R2, R3] talk about intention-conditioned trajectory prediction for autonomous vehicles. Apart from the data the methods are applied to, the architectures can be applicable to, and are relevant for, the problem being addressed here. Crucially, the DROGON paper defines intention explicitly (more on this in Weakness 2. below).*
>
> Thank you for pointing us to references [R1,R2,R3]. We cited them accordingly in the paper but refrained from going into details for the following reasons:
>
> For example, [R3] is an extension to PECNet (Mangalam et al., 2020), which is already covered, and its extension relative to PECNet do not themeselves seem to relate to our contributions.  A quantitative comparison is likewise infeasible since their method uses action labels for defining short-term intents (and there is no publically available source code). As also done in two of our baselines Weak-Sup (Zhan et al., 2019) and DAG-Net (Monti et al., 2021) reference [R1] defines intentions as future agent positions on a discretized position space. A key difference between the two application cases, however, is how these intents are computed. While [R1] (and similar work for vehicles and pedestrians) compute agent goals/intents based on fixed time horizons, Weak-Sup and DAG-Net define them as stationary points where agent speed falls below a predefined threshold to address the inherent complexity of multiagent sports data. The use of fixed time windows such as endpoints (as employed in the vehicles and pedestrain papers) is too rigid for highly interactive and nonlinear sports datasets and consequentially cannot be inferred with satisfacory precision. Empirically, an indication of this can be found in Makansi et al. (2022), where PECNet performs significantly worse on basketball data than its non goal-conditioned competitors (while achieving SOTA performance for non-interactive pedestrian datasets). Since [R1], in turn, performs over 70% worse than PECNet on e.g. the SDD benchmark, a computational comparison or detailed discussion of [R1] and other endpoint-conditioned trajectory forecasting frameworks is out of scope. Especially because our work does not explicitly aim to encode agents' intents (more on this when we discuss your concerns about our claims).
>
> *[R4] from the Neurips I Can't Believe It's Not Better Workshop explicitly deals with graph conditional variational methods for multi-agent trajectory prediction. The results in that paper are very relevant for this research area and should be included.*
>
> The paper investigates wether graph conditional variational methods for predicting multi-agent trajectories and comes to the conclusion that they are often too powerful and simplified methods (e.g., MDNs) should be preferred. However, practically all SOTA papers in the multiagent realm use some form of conditional variational autoencoder as a methodological backbone in combination with graph encoding strategies to capture agent-agent interactions (cf. almost all cited papers in the related work section). The authors of [R4] also note that the results are not conclusive (as obtained on the basis of a single data set) and countless more recent papers provide contrary evidence. Similarly, our results show clearly the usefulness of latent variables for trajectory prediction tasks (cf., e.g., Appendix "Continuous Latent Variable") and many aspects of our conceptual, algorithmic and empirical contributions would not be possible without taking a VAE approach. Nevertheless we should have cited the paper right away and made up for this now in the revision.

---

> > ### Author Response · Authors · 2022-11-18
> > **Response to Reviewer Gms6 2**
> >
> > *Graph and other stochastic methods that encode social interactions between agents have been long applied to trajectory and behavior forecasating problems. [R5] explicitly incorporates a spatiotemporal graph for incorporating social interactions between agents.*
> >
> > [R5] is fundamentally different from our work in terms of interaction strategy and problem formulation. We sketch the differences from both perspectives below and also added a corresponding short summary in the paper (cf. related work section):
> >
> > **Problem** **Formulation**
> >
> > [R5] adresses human trajectory forecasting as a deterministic regression problem by minimizing the negative log-likelihood and assuming bi-variate Gaussian output distributions. While using unimodal Gaussians may be sufficient for pedestrian datasets with rather linear trajectories, they fall short for multimodal distributions like in team sports (see, e.g., [A, B, C]). SOTA trajectory prediction methods use deep generative models such as a conditional variational autoencoders as a methodological backbone. Since our formulation belongs to the latter category, we provide an overview over such methods in the  related work section.  Thus, to ensure fair comparisons with [R5], we would need to modify our experimental setup by, for example, using the most likely single output of our model and abstracting from min ADE/FDE metrics. Although this is feasible in principle, we do not feel that this would provide new insights for the research community or our paper as many works already provide comparisons between these model classes (see, e.g., [B,C,D]).
> >
> > **Explicit Interaction Modeling**
> >
> >  [R5] models both the temporal and social dimension using graph convolutional networks and define initial edge representations with simple distance-based heuristics. There are numerous works that use similar strategies to [R5] for modeling multi-agent interactions, but are formulated as conditional variational autoencoders and are thus more relevant to us (cf. first paragraph in our related work section). However, all these techniques model interactions *implicitly* since deep graph approaches extract spatiotemporal representations by aggergating messages along edges in each network layer. On the other hand, we propose a representation learning approach to uncover such implicit knowledge explicitly in form of agent and interaction categories. The most relevant related work in this context is discussed in the second paragraph of Section 5. We updated our manuscript to emphasize these differences.
> >
> > [A] Brefeld, Ulf, Jan Lasek, and Sebastian Mair. "Probabilistic movement models and zones of control." *Machine Learning* 108.1 (2019): 127-147.
> >
> > [B] Yeh, Raymond A., et al. "Diverse generation for multi-agent sports games." *Proceedings of the IEEE/CVF Conference on Computer Vision and Pattern Recognition*. 2019.
> >
> > [C] Zhan, Eric, et al. "Generating Multi-Agent Trajectories using Programmatic Weak Supervision." *International Conference on Learning Representations*. 2019.
> >
> > [D] Salzmann, Tim, et al. "Trajectron++: Dynamically-feasible trajectory forecasting with heterogeneous data." *European Conference on Computer Vision*. Springer, Cham, 2020.
> >
> >  *[R6] more recently explicitly takes a meta-learning approach for modeling the dynamics unique to a group for probabilistic forecasting. A sports team is a group, and if each team is viewed as having unique social dynamics resulting from the team's strategy then [R6]'s core modeling idea is directly applicable.*
> >
> > Thank you, that is an interesting future line of research and we added this as such in the paper (cf. Section 6). Modeling unique team strategies is certainly a desired quantity but also requires a sufficient amount of data from all involved teams as behavior and strategies are apdated from game to game and only a large number of observations will give rise to consistent unique behavior across different opponents. Apart from unclear evaluation of such charactersistic traits, we simply lack data for such an undertaking and it was also not the goal of the paper.

---

> > > ### Author Response · Authors · 2022-11-18
> > > **Response to Reviewer Gms6 3**
> > >
> > >  *Their modeling of social influence of other agents is also permutation invariant, a limitation this paper claims about existing methods.*
> > >
> > > The sentence about about permutation-invariance in our background section refers to the adjustments required for appropriately modeling multiagent data. It implies that if we were to use Eq. (1) directly (which at this point only captures temporal dependencies), independence over the agent dimension is required to achieve permutation-invariance which in turn is a prerequisite for processing sequences of unordered collections like multi-agent trajectories. This independence assumption is trivially inappropriate for interactive systems, so work in this area (besides [R6], almost all cited trajectory papers in our manuscript) proposes sensitive solutions (as discussed above, usually in the form of some graph encoding strategy) to capture agent-agent interactions while maintaining permutation-invariant functions. In fact, it is one of the key reasons why graph neural networks are employed in the first place.  As such, it merely acts as a "step-by-step” explanation for summarizing motivations of previous works (such as GVRNN) on using graph-based tools.
> > >
> > > *there is no reasonable way to evaluate whether the proposed method actually models intentions. The intention-conditioned trajectory works I've mentioned talk about intention over long- and short- time horizons, where e.g. the former is in terms of goal destinations. Here the paper is talking about team sports with player intentions but simply states that this results from communication. What does intention mean here? Also, the paper claims to model causal relationships, but I can't see any explicit causal factors modeled or learned in the graph structure. There might be other exogenous variables explaining trajectory behavior.*
> > >
> > > That's right. We do not aim to explicitly model agent intentions. In fact, we mentioned intention only twice in the entire manuscript as motivating examples (as common in the VAE literature) for latent design choices. We quote directly from the originally submitted version:
> > >
> > > - “However, being the only causal factors specified, the proposed frameworks neglect other potential latent characteristics not originating in mere interactive categories but equally affecting multimodal agent behavior, such as agent intents” (section 1, end of third paragraph).
> > > - “Multimodal agent behavior is affected not only by the structural state captured in G_t, but also by agent intents, in turn contingent on the inferred communication mechanisms and agent roles. Intuitively, each realization z_t thus encodes latent factors of variation in the trajectories not represented by the inferred causal graphs, yet may vary as a function of G_t.” (section 3.1.)
> > >
> > > As described in section 3.1., we inherit the latent concepts z_t from GVRNNs since our proposed latent graph may not be sufficiently informative to encapsulate all sources of uncertainty. Introducing z_t thus merely aims to improve models’ generative capacity via capturing these remaining factors. A logical candidate for such a factor would be, for example, agent intents (defined in a broader intuitive context). We updated the paragraph describing the latent variables to remove the ambiguity this term introduced and emphasize that it can contain arbitrary properties for modeling model uncertainty. Our empirical results confirm that this is successful (cf. Appendix C.5).
> > >
> > > (cont.)

---

> > > > ### Author Response · Authors · 2022-11-18
> > > > **Response to Reviewer Gms6 4**
> > > >
> > > > Instead, the clear focus throughout our work is to maximize the information that G_t contains about characteristics relating to social factors through our latent space structuring and model setup. From Eq (9) and Eq (10), the predicted agent and interaction types define the parameters of all subsequent graph networks that infer variables x_t and z_t. Thus, the realized values in G_t affect movement predictions both directly (through the decoder) and indirectly (throuph prior and encoder): it defines the actual information that is collected over the agent dimension for carrying out those predictions. Since we minimize a reconstruction metric, this ensures some notion of social causality in G_t. Our empirical results confirm that his is successful on generative tasks and unrelated downstream applications:
> > > >
> > > > - Figure 3 (left) provides GVRNN comparisons with a modified DIA version for different observation and prediction periods. Since the achieved performance gains originate from the proposed graph structure, we can conclude that G_t effectively both eliminates interfering social signals and discriminates between different agent and interaction types so that future agent movements can be captured more accurately.
> > > > - Figure 3 (right) and Appendix C.4 show declining performances when social representation capacity exceeds expected social complexity. While it may not be conclusive, it does provide some indication that G_t coincides with ground truth factors of interest.
> > > > - Qualitative evaluations in Section 4.3 and Appendix C.6 indicate that the model does learn an interpretable sematic structure.
> > > > - In Section 4.2. (paragraph “Auxiliary Task”), we provide quantitative evidence for the model capturing some notion of importance between agents that can be leveraged effectively in downstream applications.
> > > >
> > > > At R4's request, we conducted additional experiments:
> > > >
> > > > - We compare DIA and GVRNN generative model performance for varying data fractions of 5%, 10%, 20% and 50% of the training data. This set of experiments thus aims at investigating whether the learned structure translates into data efficiency by, for example, governing attention to important pieces of data. See Figure below (or Appendix C.3 in the revised manuscript).  Impressively, our model still *significantly* outperforms the full GVRNN baseline on both error metrics using only 5% (!) of the available training data.
> > > >
> > > > **Note:** The Figure can be also accessed through the following anonymous link (and Appendix C.3 in revised manuscript): https://ibb.co/cc5HM5y
> > > >
> > > > - The table below (Table 3 in the revised paper) aims to directly compare the importance of structure vs. larger latent capacity. Clearly, larger decoders/encoder/priors yield degenerate results compared to the initial (best) GVRNN configuration. Furthermore, the DIA comparisons highlight the significant importance of enforcing a graph structure since the only difference between both model configurations is our interaction learning strategy.
> > > >
> > > >
> > > >     |  | minADE | minFDE | # Parameters |
> > > >     | --- | --- | --- | --- |
> > > >     | GVRNN (best) | 9.73 | 15.80 | 355588 |
> > > >     | GVRNN (x2 width) | 9.91 | 16.12 | 733700 |
> > > >     | GVRNN (x3 width) | 9.94 | 16.20 | 1647364 |
> > > >     | DIA (no structure) | 10.27 | 15.17 | 400836 |
> > > >     | DIA | 8.29 | 12.05 | 421768 |
> > > >
> > > > ### Notation
> > > >
> > > > *There are a few notational errors. For instance, the variable used for the sequence cannot be the same as the individual elements: x<t=[x1,...]. See [R4] for this.*
> > > >
> > > > We note that [R4] conducts experiments with both sequential 1D and sequential multi-agent data; thus, such a distinction is required. However, we explicitly define x_{≤T} as sets of sequences.
> > > >
> > > > *In many places there exist grammatical errors and incomplete sentences. Please do a pass to fix these.*
> > > >
> > > > Thank you for pointing this out. We proofread the original version and corrected all flaws in the revision.

---

### Official Review · Reviewer_JaLg · 2022-10-31

**Confidence:** 4
**Correctness:** 3
**Technical Novelty And Significance:** 3
**Empirical Novelty And Significance:** 3
**Recommendation:** 6

**Clarity, Quality, Novelty And Reproducibility:**

Quality: The paper is interesting and novel.

Clarity: the paper is clear and well written.

Originality: To the best of my knowledge, the proposed idea is novel.

**Strength And Weaknesses:**

Strengths,
* The proposed method is interesting and novel, in particular the graph based latent space.
* The proposed method performs well on soccer data outperforming several state of the art methods.
* The paper is well written and easy to understand.

Weakness,
* Recent works on trajectory prediction in a social context are not discussed: "Euro-pvi: Pedestrian vehicle interactions in dense urban centers, CVPR 2021" which also develops a  variational auto-encoder based joint inference model that learns an expressive multi-modal shared latent space across agents. A discussion and quantitative comparison is desirable.
* Methods developed for pedestrian trajectory prediction are in general applicable to the target soccer data as these methods also aim to predict trajectories while modelling the effect of social interactions. e.g. Trajectron++, ECCV 2020. A quantitative comparison to such pedestrian trajectory prediction methods are desirable, including a discussion on the advantages/disadvantages of such models for sports data.
* Sampling speeds: Inference with a complex graph based latent space would be slow. A quantitative comparison of sampling speeds across state of the art models would be desirable.
* Datasets: It would also be interesting to benchmark the proposed method on pedestrian trajectory prediction datasets e.g. Stanford Drone, Waymo Open, nuScenes to illustrate the differences and challenges involved i sports data vs pedestrian data.

**Summary Of The Paper:**

This paper proposes a method for trajectory prediction in a social context. The proposed model is a variational auto-encoder with a graph based latent space. The graph based latent space is used to capture the effect of social interactions. Evaluation is performed on soccer data which contains trajectories of soccer players and the ball.

**Summary Of The Review:**

Although the paper is interesting and novel, a detailed comparison with methods from the very related area of pedestrian trajectory prediction is desirable. Moreover, a detailed analysis of sampling speeds would be welcome.

---

> ### Author Response · Authors · 2022-11-18
> **Response to Reviewer JaLg 1**
>
> We would like to thank you for your praise and constructive suggestions for improving the paper. We hope that our response as well as the corresponding paper updates alleviate your remaining concerns.
>
> *Recent works on trajectory prediction in a social context are not discussed: "Euro-pvi: Pedestrian vehicle interactions in dense  urban centers, CVPR 2021" which also develops a variational auto-encoder based joint inference model that learns an expressive multi-modal shared latent space across agents. A discussion and quantitative comparison is desirable.*
>
> Thank you for bringing this work to our notice; this paper is indeed relevant and should be mentioned. As suggested, we also performed quantitative comparisons, shown below (and in the updated manuscript).
>
> | Method | minADE/FDE (long-term) | minADE/FDE (short-term) |
> | --- | --- | --- |
> | Joint- ⁍-cVAE | 10.64/14.47 | 4.03/6.56 |
> | Ours | 8.29/12.05 | 2.20/4.51 |
>
> As can be seen, Joint- $\beta$-CVAE performs significantly worse compared to our method. To some extent, this empirically confirms our original finding that methods with temporally global latent variables do not sufficiently capture multiagent dynamics in sports environments. We added a discussion to the manuscript (more on this below).
>
> *Methods developed for pedestrian trajectory prediction are in general applicable to the target soccer data as these methods also aim to predict trajectories while modelling the effect of social interactions. e.g. Trajectron++, ECCV 2020. A quantitative comparison to such pedestrian trajectory prediction methods are desirable, including a discussion on the advantages/disadvantages of such models for sports data*.
>
> We agree that comparisons/discussions targeting methodological transferability from other domains would provide additional valuable insights to the community. Generally, we observed that SOTA methods for urban environments are formalized as some form of conditional variational autoencoder that aggregates latent information across time (such as Trajectron++, Joint- $\beta$-cVAE, or GRIN, one of our baselines that also has been tested on pedestrian data). Though it may be premature to arrive at a definite conclusion, the comparatively dismal performance of such baselines in our results tends to suggest limited transferability. We hypothesize that this is due to lower inherent social interaction patterns within these standard pedestrian datasets (clf. Makansi et al., 2022; more on this below). We extended our discussion in the manuscript on this topic in Section 4.1. (first paragraph) with the newly integrated results from above. (We note that Joint- $\beta$-cVAE performs better in direct comparison with Trajectron++ and we therefore refrain from an additional comparison with the latter.)
>
> *Datasets: It would also be interesting to benchmark the proposed method on pedestrian trajectory prediction datasets e.g. Stanford Drone, Waymo Open, nuScenes to illustrate the differences and challenges involved sports data vs pedestrian data.*
>
> This is a highly interesting discussion, but has already been examined at paper length in Makansi et al. (2022). In fact, the results of this study were an important reason that ultimately led to choosing our data: our contribution aims to explicitly uncover dynamic interaction patterns, but such social patterns are scarce in pedestrian datasets in that empirical results can be largely explained by past agent observations. Thus, additional generative comparisons on these benchmarks would be uninformative for our goal.
>
> *Sampling speeds: Inference with a complex graph based latent space would be slow. A quantitative comparison of sampling speeds across state of the art models would be desirable.*
>
> We added information about parameter size and sampling speed of the best performing models. The Table below shows the parameter size and sampling speed expressed as averaging over mini-batches for modeling offensive players in basketball.
>
> |  | Parameter count | Sampling speed |
> | --- | --- | --- |
> | GVRNN | 355588 | 0.0704 |
> | DAG-Net | 184040 | 1.4581 |
> | DIA (Ours) | 421768 | 0.0931  |
>
> While our model is slower than GVRNN, we can significantly reduce runtime, as evidenced by additional experiments we conducted on data efficiency:
>
> **Note**: The Figure can be also accessed through the following anonymous link (and Appendix C.3 in revised manuscript):
> https://ibb.co/cc5HM5y
>
>
> As can be seen, our model still *significantly* outperforms the full **GVRNN baseline on both error metrics using only 5% (!) of the available training data. Thus, we can reduce runtime by at least 10x compared to GVRNN while maintaing superior results.

---

### Decision · Program_Chairs · 2023-01-20

**Decision:**

Reject

**Justification For Why Not Higher Score:**

While the proposed approach is interesting, the paper could do a better job at positioning itself with respect to the existing literature, and comparing against these approaches.

**Justification For Why Not Lower Score:**

N/A

**Metareview: Summary, Strengths And Weaknesses:**

The reviewers agree that this is an interesting application with a rich area of research. In theory, the proposed method could be applied to a large number of different settings.

However, there are concerns regarding the novelty of the approach, in the sense that it could be better positioned with regards to existing literature on e.g. pedestrian tracking. Relatedly, the reviewers feel that the additional complexity of the proposed approach is not completely justified by the current experiments.